# Is it possible to reconstruct an accurate cell lineage using CRISPR recorders?

Irepan Salvador-Martínez[1†], Marco Grillo[2,3†], Michalis Averof[2,3*], Maximilian J Telford[1*]

[1]Centre for Life's Origins and Evolution, Department of Genetics Evolution and Environment, University College London, London, United Kingdom; [2]Institut de Génomique Fonctionnelle de Lyon (IGFL), École Normale Supérieure de Lyon, Lyon, France; [3]Centre National de la Recherche Scientifique (CNRS), Paris, France

**Abstract** Cell lineages provide the framework for understanding how cell fates are decided during development. Describing cell lineages in most organisms is challenging; even a fruit fly larva has ~50,000 cells and a small mammal has >1 billion cells. Recently, the idea of applying CRISPR to induce mutations during development, to be used as heritable markers for lineage reconstruction, has been proposed by several groups. While an attractive idea, its practical value depends on the accuracy of the cell lineages that can be generated. Here, we use computer simulations to estimate the performance of these approaches under different conditions. We incorporate empirical data on CRISPR-induced mutation frequencies in *Drosophila*. We show significant impacts from multiple biological and technical parameters - variable cell division rates, skewed mutational outcomes, target dropouts and different sequencing strategies. Our approach reveals the limitations of published CRISPR recorders, and indicates how future implementations can be optimised.
**Editorial note:** This article has been through an editorial process in which the authors decide how to respond to the issues raised during peer review. The Reviewing Editor's assessment is that all the issues have been addressed (see decision letter).
DOI: https://doi.org/10.7554/eLife.40292.001

*For correspondence:
michalis.averof@ens-lyon.fr (MA);
m.telford@ucl.ac.uk (MJT)

†These authors contributed equally to this work

Competing interests: The authors declare that no competing interests exist.

## Introduction

Starting from a single cell - the fertilised egg - multicellular organisms undergo repeated rounds of cell division to produce the adult form. The divisions that generate these adult cells constitute a genealogical tree with the fertilised egg at its root and each adult cell as a terminal branch. Knowing the cell lineage that produces a fully developed organism from a single cell provides the framework for understanding when, where and how cell fate decisions are made.

Obtaining high resolution (single-cell level) lineages is a challenging task that has been solved only in animals with relatively few cells, such as the nematode *Caenorhabditis elegans*: its complete lineage (~1000 cells) was deduced by painstaking observation of each cell division under the microscope. This approach is impossible in larger animals, in which most cells are inaccessible to microscopy and their number becomes quickly unmanageable. The 16 rounds of cell division required to produce a hatched *Drosophila* larva, for example, result in about 50,000 cells (*Lehner et al., 2001*) and further rounds of division produce an adult with approximately $10^6$ cells. The bodies of mice and humans consist of $10^{10}$ to $10^{14}$ cells respectively (*Sender et al., 2016*).

Recently it was proposed that naturally occurring somatic mutations, which accumulate in cells during the lifetime of an organism, could be used as lineage markers to reconstruct its entire cell lineage (*Frumkin et al., 2005*; *Salipante and Horwitz, 2006*). This is directly analogous to the use of heritable mutations, accumulating through time, to reconstruct a species phylogeny. While this

approach is theoretically possible (*Frumkin et al., 2005*), it is nevertheless limited by the enormous challenge of detecting these rare mutations within the genomes of individual cells.

As a solution to the problem of reading the mutations, several recent papers have explored the idea of using CRISPR-induced somatic mutations, targeted to artificial sequences inserted as transgenes into the genome (termed 'CRISPR recorders') (*McKenna et al., 2016*; *Frieda et al., 2017*; *Junker et al., 2016*; *Kalhor et al., 2018*; *Perli et al., 2016*; *Alemany et al., 2018*; *Schmidt et al., 2017*; *Raj et al., 2018*; *Attardi et al., 2018*; *Spanjaard et al., 2018*; *Junker et al., 2016*). The recorders consist of arrays of CRISPR target sites, targeted by their cognate sgRNAs and Cas9 during development. Starting in early embryogenesis, CRISPR-induced mutations occur stochastically at these target sites, in each cell of the body, and these mutations are stably inherited by the progeny of these cells. In most cases, the mutation destroys the match between target and sgRNA meaning a mutated target is immune to further change. At the end of development only the recorder sequence has to be read rather than the whole genome; the accumulated mutations can then be used as phylogenetic characters allowing the reconstruction of a tree of relationships between all cells (*Figure 1*).

The basic principle of recorder-based lineage tree reconstruction is easy to grasp. What is far less clear is whether the lineages produced by these methods are accurate enough for us to draw meaningful conclusions from them. Of course the required accuracy will depend on the intended use of the lineage, but to date few studies have considered how accurate the lineages produced might be (*Frieda et al., 2017*; *Schmidt et al., 2017*; *Spanjaard et al., 2018*).

The ideal way of assessing the accuracy of these techniques would be to compare the real cell lineage of an organism against the lineage inferred by the recorder (*Schmidt et al., 2017*). This is difficult to implement in practice, however, because in most cases the real cell lineages are unknown.

We have taken the alternative approach of computationally simulating the processes of cell division and accumulation of mutations in a recorder and then comparing the lineage inferred from the recorder to the known *in silico* reference tree. We have used this approach to estimate the accuracy of lineage reconstruction in different situations (type and complexity of recorder, mutation rates, cell

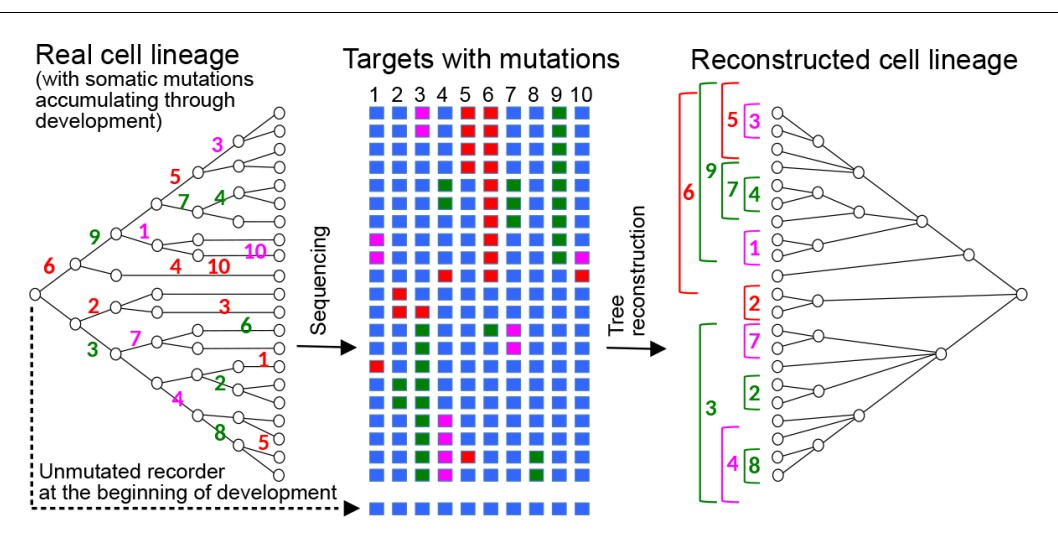

**Figure 1.** Reconstructing cell lineages using CRISPR-induced somatic mutations. Left: Development begins with a zygote carrying in its genome a lineage recorder composed of a series of CRISPR targets (blue boxes). During subsequent cell divisions, any target of the recorder can be cleaved by Cas9 in any cell, leaving a specific mutational signature on the target which will be inherited by all the descendants of the cell. Numbers represent the the cleaved target in the recorder and its mutational signature is represented by a colour. Middle: At the end of development, the recorder of every cell is sequenced, recovering the pattern of accumulated mutations in each of the targets (coloured boxes). Right: The pattern of mutations is used to reconstruct the cell lineage, in a similar way to how a phylogenetic tree is inferred from the sequences of homologous genes.
DOI: https://doi.org/10.7554/eLife.40292.002

lineage depth, etc.), taking into account empirical measures of mutation rates and frequencies of different mutational outcomes derived from *in vivo* experimental data from *Drosophila melanogaster*. While some previous studies used simulations to evaluate the reconstruction of small cell lineages, no study has attempted this on cell lineages of tens of thousands of cells (*Frieda et al., 2017*; *Schmidt et al., 2017*).

Different designs of CRISPR recorders have been implemented, including recorders that register point mutations on arrays of barcoded targets (GESTALT; *McKenna et al., 2016*; *Raj et al., 2018*), ones that rely on 'collapsing' target arrays through deletions (MEMOIR; *Frieda et al., 2017*), recorders that target identical target sites located on separate transgenes (ScarTrace and LINNAEUS; *Junker et al., 2016*; *Schmidt et al., 2017*; *Attardi et al., 2018*; *Alemany et al., 2018*; *Spanjaard et al., 2018*) and ones that target the CRISPR sgRNA itself (*Kalhor et al., 2018*; *Perli et al., 2016*). In this work we have simulated the behaviour of the first two types of recorders, but the insights that we have gained should apply to all types of recorders. Ultimately, these simulations will help us to establish a set of criteria for the optimal design of CRISPR-based lineage recorders, as well as to understand the limitations of these techniques when addressing real biological questions.

To assess the power of CRISPR-based lineage recorders in cell lineage reconstruction, we focus on the conditions required to reconstruct a cell lineage of ~65,000 cells. This roughly corresponds to the size of the cell lineage of a *Drosophila* first instar larva, of a pharyngula stage zebrafish embryo, or a stage E8.0 mouse embryo (*Lehner et al., 2001*; *Kane, 1999*; *Kojima et al., 2014*, respectively).

## Results

### General description of the simulations

In our simulations, a cell is implemented as a vector of $m$ targets. We begin each simulation with one cell, representing the fertilised egg, that has all its targets in an unmutated state. The initial cell then undergoes a series of cell divisions ($d$), growing into a population of $N$ cells, where $N = 2^d$. Following each cell division, each unmutated target can mutate (with a given probability $\mu_d$) to one of several possible mutated states. Once a target is mutated, it can no longer change, either to revert to the unmutated state or to transit to a new state (*Figure 2A*). We expect the frequency of reversions to be negligible as even single nucleotide changes result in a large decrease in mutation rate.

A unique label was given to each cell during the simulation. The sequence of simulated cell divisions were recorded in the form of a tree, whose topology describes the lineage relationships between all cells (the 'reference tree'). At the end of the simulation, we randomly sampled a number of cells and used the pattern of mutations accumulated in those cells to infer their cell lineage (the 'inferred tree') using the Neighbor-Joining method (*Saitou and Nei, 1987*) (for a comparison with parsimony see *Figure 2—figure supplement 1*).

The accuracy of lineage reconstruction of each simulation was determined by comparing the inferred tree with the reference tree using a measure derived from the Robinson-Foulds algorithm (*Robinson and Foulds, 1981*), which calculates the percentage of splits in the reference tree that are precisely recovered in the inferred tree (*Figure 2B*). If the inferred tree is identical to the reference tree, the accuracy is 100%. This provides a strict measure of the global accuracy of the inferred lineage tree. The accuracy of each lineage reconstruction was estimated as the mean accuracy of 10 subsamples of 1000 cells.

### Impact of mutation rate on the accuracy of cell lineage reconstruction

We simulated a lineage with a depth of 16 cell divisions ($d = 16$), yielding 65,536 cells ($2^{16}$). To determine the effect of varying the mutation rate on the accuracy of lineage reconstruction, we performed simulations with a recorder carrying 100 targets ($m = 100$) and a rate of mutation $\mu_d$ varying from 0.01 to 0.3 mutations per cell division per target. In parallel, we tested how the diversity of mutational outcomes (number of distinct mutated states) at each target could influence the accuracy of lineage reconstruction, by varying the number of possible mutational outcomes at each target between 2 and 32. In each case the different mutational outcomes were considered to be equiprobable.

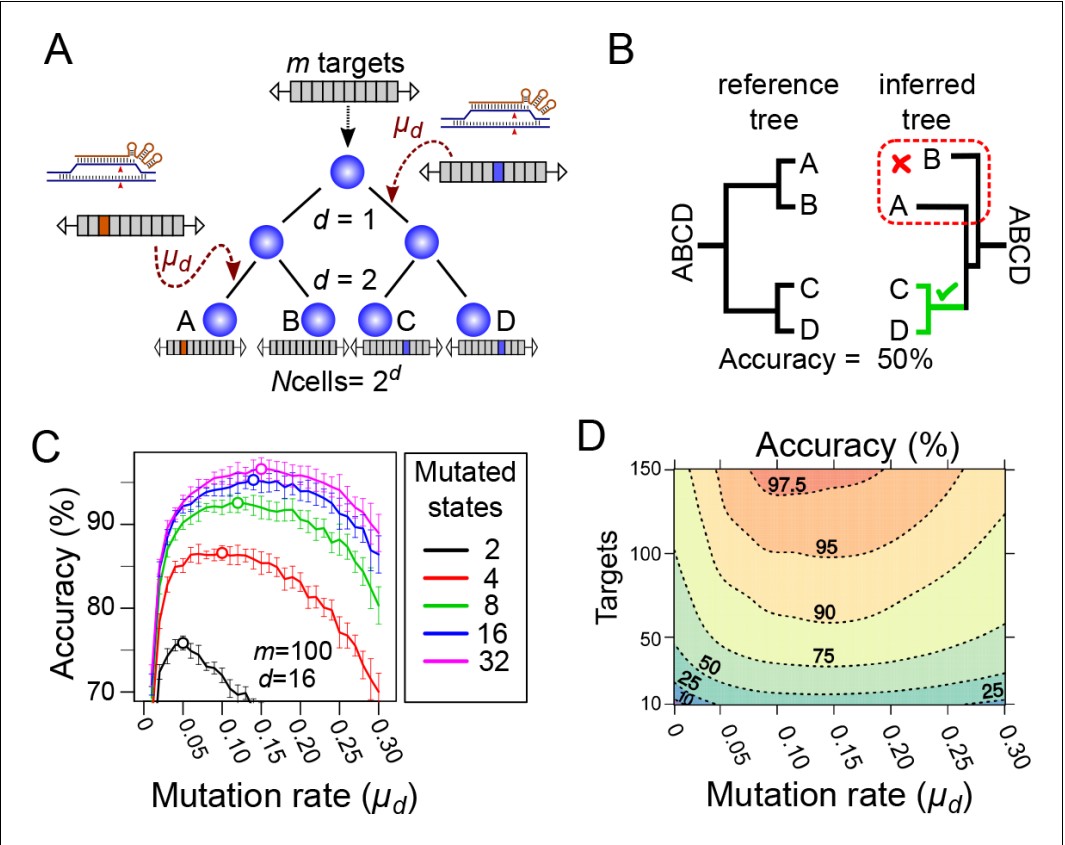

**Figure 2.** Computational simulation of CRISPR recorders (**A**). In our simulations, an initial cell with multiple CRISPR targets ($m$) yields $N$ cells after a given number of cell divisions ($d$). The recorder accumulates independent CRISPR-induced mutations with a probability, in each target, of $\mu_d$ per cell division; the mutations are inherited in subsequent cell divisions. The pattern of mutations accumulated in each cell is used to infer the lineage tree. (**B**) The accuracy of lineage reconstruction was determined by comparing the inferred tree with the reference tree using using a measure related to the Robinson Foulds method. The unmutated state of the recorder was used to root the tree. (**C**) Accuracy of lineage reconstruction with a recorder of 100 CRISPR targets after 16 cell divisions (yielding 65,536 cells) over a range of mutation rates. Each line represents the mean accuracy (10 simulations) for simulations resulting in different numbers of equiprobable mutated states. The optimal mutation rate for each number of mutated states is indicated with an open circle. Vertical lines represent 95% confidence intervals. (**D**) Accuracy of lineage reconstruction for different mutation rates and numbers of CRISPR targets. Mutations were set to result in 16 equiprobable mutated states. Dashed lines represent different accuracy thresholds (levelplot) after a LOESS regression. For each parameter combination, we plot the mean accuracy of 10 simulations after 16 cell divisions.

DOI: https://doi.org/10.7554/eLife.40292.003

The following figure supplement is available for figure 2:

**Figure supplement 1.** Comparing the performance of Neighbor Joining and Maximum Parsimony in lineage reconstruction.

DOI: https://doi.org/10.7554/eLife.40292.004

As expected, these simulations show that the mutation rate and the diversity of mutations have a strong effect on the accuracy of cell lineage reconstruction (*Figure 2C*). A low diversity of possible mutational outcomes gives poorer results than a higher diversity. Mutation rates show a broad optimum between between 0.05 and 0.2 mutations per cell division per target; under these rates, 56–97% of target sites are mutated after 16 cell divisions. Lower mutation rates lead to more targets having no mutations, thus contributing no information for reconstructing the cell lineage. Higher mutation rates lead to most targets being mutated during the early cell divisions, leaving few targets available for recording later events.

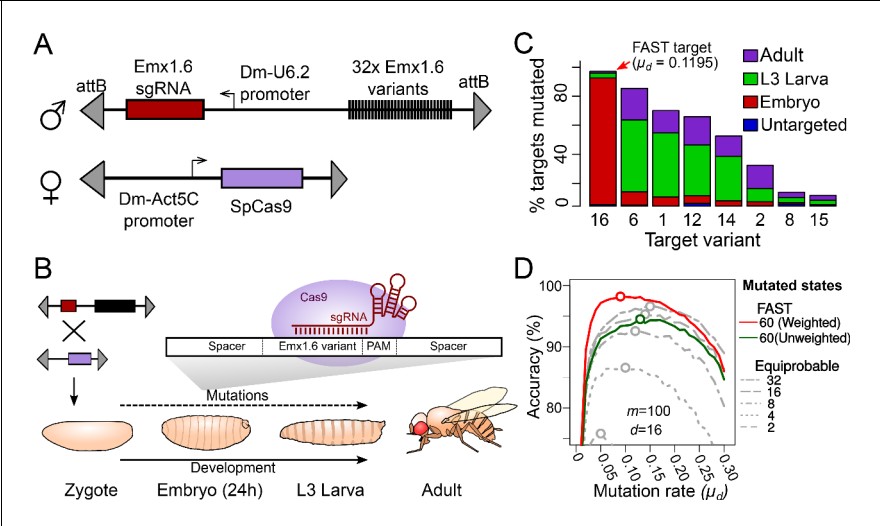

**Figure 3.** Tuning the mutation rate of a CRISPR recorder in vivo. (**A**) CRISPR recorder designed to test the mutation rates of 32 variants of the Emx1.6 target in *Drosophila*. The recorder consists of two transgenic constructs brought together by genetic crosses. The first construct carries an array of the 32 target variants and a transgene expressing the Emx1.6 sgRNA under the constitutive *Drosophila* U6.2 promoter (*Port et al., 2014*). The second construct expresses the *Streptococcus pyogenes* Cas9 gene under the constitutive *Drosophila* Act5C promoter (*Port et al., 2014*). (**B**) Double heterozygotes carrying both constructs were collected at embryonic, late larval (L3) and adult stages and analysed for mutations in the target array by PCR amplification and sequencing of the recorder. (**C**) Proportion of targets mutated at different stages, for the eight most efficient target variants. 'Untargeted' represents background mutations or sequencing errors observed in the absence of the Cas9 transgene. (**D**) Estimates of cell lineage accuracy from computer simulations (as in *Figure 2C*) using the mutational outcomes observed in vivo on the FAST target.

DOI: https://doi.org/10.7554/eLife.40292.005

In practice, CRISPR activity generates a range of mutations (mostly small deletions or insertions) at varying frequencies. We have measured the actual rates and diversity of CRISPR-induced mutations in vivo (see below) and used these empirical data in our subsequent simulations.

## Tuning the mutation rate of a CRISPR recorder in vivo

Our simulations show that specific mutation rates must be achieved experimentally in order to optimise cell lineage reconstruction. There are several ways to vary CRISPR mutation rates in vivo, including the use of different sgRNA:target pairs, varying the expression levels of sgRNA and Cas9, and using variants of sgRNA or Cas9 that influence their stability or activity. We chose to adjust the mutation rate by altering the target sequence in order to introduce mismatches in the sgRNA:target pairing (similar to the strategy used on cultured cells by *McKenna et al., 2016*); this is known to reduce the targeting efficiency (*Hsu et al., 2013*; *Fu et al., 2014*). We have measured the mutation rate of a series of different variants of a CRISPR target to find those with the optimum rates for cell lineage reconstruction of the *Drosophila* embryo.

We took advantage of a previous study by *Hsu et al. (2013)* who analysed the effects of sgRNA:target pairing mismatches on the efficiency of targeting a section of the human *EMX1* gene. Based on this study, we selected 32 variants of the Emx1.6 target (*Hsu et al., 2013*), including the wild-type sequence and 31 variants with single- or double-nucleotide changes at different positions within the target sequence (*Supplementary file 1—Table 1*). To compare the mutation rates of the 32 targets, we designed and synthesised a single construct that carries all 32 variants in tandem. In the same construct, we incorporated a transgene constitutively expressing the Emx1.6 sgRNA under the *Drosophila* U6.2 promoter (*Port et al., 2014*) (*Figure 3A*). We generated transgenic *Drosophila* lines carrying a single copy of this construct at the 37B7 locus, using $\phi$C31-mediated integration (*Bateman et al., 2006*).

Males carrying the Emx1.6 sgRNA and the target array were crossed with virgin females carrying a constitutively expressed Cas9 transgene (Actin::Cas9, *Port et al., 2014*) to generate progeny carrying a single copy of the CRISPR target array, the sgRNA and the Cas9 transgene. We collected these progeny at different developmental stages (end of embryogenesis, late L3 larvae, newly

eclosed adults) to assess the number of mutations that had accumulated in each of the 32 target variants at these three stages (*Figure 3B*).

We pooled individuals collected at each of the three chosen developmental stages, performed PCR on genomic DNA and used high throughput sequencing to characterise the mutated targets. In individual animals, mutational frequencies are influenced both by the probability of each mutational outcome of CRISPR and by the clonal expansion of cells that carry each mutation. Since we have analysed populations of individuals and expect no systematic clonal biases associated with specific mutations in CRISPR recorders, we expect that our estimates of mutational frequencies largely reflect the frequencies of CRISPR-induced mutations on our target.

Our results confirm that, by employing different target variants, we can achieve widely different rates of mutation. As expected, the target that has perfect complementarity with the Emx1.6 sgRNA (target 16, named the 'FAST' target) showed the highest mutation rate; having corrected for sequencing errors (~1% of control targets have differences due to PCR or sequencing errors, *Supplementary file 1—Table 1*) we observed that 87% of the targets carried a mutation at the end of embryogenesis (*Figure 3C*). This corresponds to a mutation rate of $\mu_d$ = 0.1195 per cell division, assuming a constant mutation rate per cell division (see later for consideration of uneven rates per cell division). The other targets showed lower mutation rates: in the six variants with the highest rates, $\mu_d$ ranged from $4 \times 10^{-4}$ (target 15) to $6 \times 10^{-2}$ (target 6) mutations per cell division (*Supplementary file 1—Table 2*).

The mutation rate of the FAST target ($\mu_d$ = 0.1195) falls within the optimal range we had estimated for reconstructing the lineage of 65,536 cell embryos, assuming a uniform rate of cell division (see *Figure 2C*). Targets with slower mutation rates would be suited for lineaging past the embryonic stages. Conversely, faster mutation rates would be optimal for lineaging embryos at earlier stages, following fewer cell divisions. Instances of rapid or unequal rates of cell division would also require faster mutation rates (see below).

## Simulating cell lineage reconstruction based on experimentally observed mutational outcomes

Thus far, our simulations assumed that the targets can mutate to a certain number of character states with equal probability. This assumption does not reflect the complexity of CRISPR mutagenesis observed *in vivo*. Our sequencing data for the 32 variants of the Emx1.6 target in *Drosophila* provide empirical measurements not only of the rate of mutation but also of the diversity of different mutational outcomes and their relative frequencies in a CRISPR recorder *in vivo*. Using these data we refined our simulations using the real set of mutational outcomes and their observed relative frequencies.

We focused on the complexity of mutational outcomes affecting the FAST target. As reported in previous studies (*van Overbeek et al., 2016*; *Allen et al., 2018*; *Chen et al., 2018*), we found that most of the mutations were located close to the Cas9 editing site. This suggests that most of the mutational information can be extracted by reading the nucleotides surrounding the editing site. Focusing on the 9 bp adjacent to the PAM sequence (*Figure 4B*) we observed >200 mutated states. The frequencies of mutations follow an exponential curve, with a few variants occurring at high frequency (*Figure 4C*), in contrast to a naive assumption of equiprobable mutational outcomes.

Using the observed distribution of these 9mers and the estimated overall $\mu_d$, we carried out 1000 simulations of the mutational process in a hypothetical construct carrying 32 identical FAST targets. We used 32 targets because we have shown that synthesising and generating transgenic flies with such a construct is feasible. For convenience, we considered 61 out of the ~200 observed states: 59 states representing the 59 most common mutations (which account for 95% of the total observed mutations; see *Figure 4C*), a 60th state with a frequency of 5% that accounts for all other outcomes combined, and the 61st state representing the unmutated target.

Using the experimentally measured distribution of mutational outcomes, the accuracy of cell lineage reconstruction is 72% (see *Figure 5B*). Rarely occurring mutations are less likely to appear independently in the same target in more than one branch of the lineage tree (an instance of homoplasy), suggesting that rare mutations are better lineage markers than more frequent mutations. To take advantage of this we introduced a weighting scheme whereby mutations are weighted in inverse proportion to their frequency of occurrence (see Materials and methods). Following this

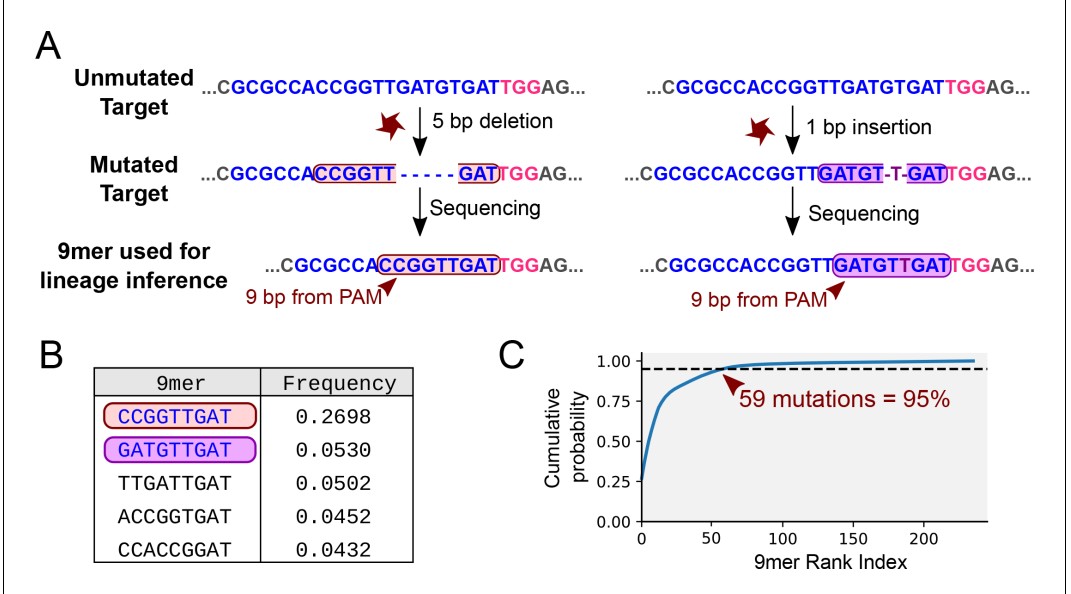

**Figure 4.** Mutational outcomes of CRISPR in vivo. (**A**) Examples of two mutational outcomes after CRISPR cleavage. The 9-nucleotide sequence located immediately upstream of the PAM (coloured box) captures most of the variation resulting from CRISPR-induced mutations (***Figure 4—figure supplement 1***). The target sequence is shown in blue, PAM sequence in pink, flanking sequence in grey. (**B**) Relative frequencies of the five most common mutational outcomes in the FAST target. (**C**) Cumulative probability of the mutational outcomes. 59 mutations account for 95% of the total number of mutations.

DOI: https://doi.org/10.7554/eLife.40292.006

The following figure supplement is available for figure 4:

**Figure supplement 1.** Accuracy of reconstruction varying the number of character states and nucleotides (Nmers) used for reconstruction.

DOI: https://doi.org/10.7554/eLife.40292.007

approach, the accuracy of lineage reconstruction using the same 61 states improved from 72% to 82% (***Figure 5B***).

## Impact of uneven rates of cell division on the accuracy of cell lineage reconstruction

So far we have assumed that the probability of mutation per available target ($\mu_d$) is the same in every cell division. This would be a reasonable assumption if all cells have a similar rate of cell division and if that rate remains constant during the course of development. In many species, however, the rate of cell division in embryogenesis varies among cells and through time. Early *Drosophila* embryos, for example, initially go through a series of 13 rapid and near-synchronous nuclear divisions to generate a uniform syncytial blastoderm (***Zalokar and Erk, 1976***); during this phase the duration of each mitotic cycle is very short (~10 min). After cellularisation at cell cycle 14, the rate of cell division in the embryo slows considerably and becomes asynchronous (***Foe, 1989***; ***Hartenstein, 1993***).

To estimate the impact of uneven rates of cell division on cell lineage reconstruction we modelled the mutation events as a Poisson process dependent on time rather than on cell divisions. A Poisson process assumes that a given event (in this case a CRISPR-induced mutation) occurs stochastically at a given rate $\mu_t$. We estimated that setting $\mu_t$ at 0.0014 mutations per site per minute would produce the observed proportion of mutated FAST targets (87%) after 24 hr of embryonic development. We set the cell division intervals to approximate those known from *Drosophila* development (see Materials and methods) and we modelled the frequency and diversity of mutational outcomes on those observed in the FAST target (see previous section). Under these conditions, we would expect the accuracy of lineage reconstruction to be considerably worse, as there will be many fewer

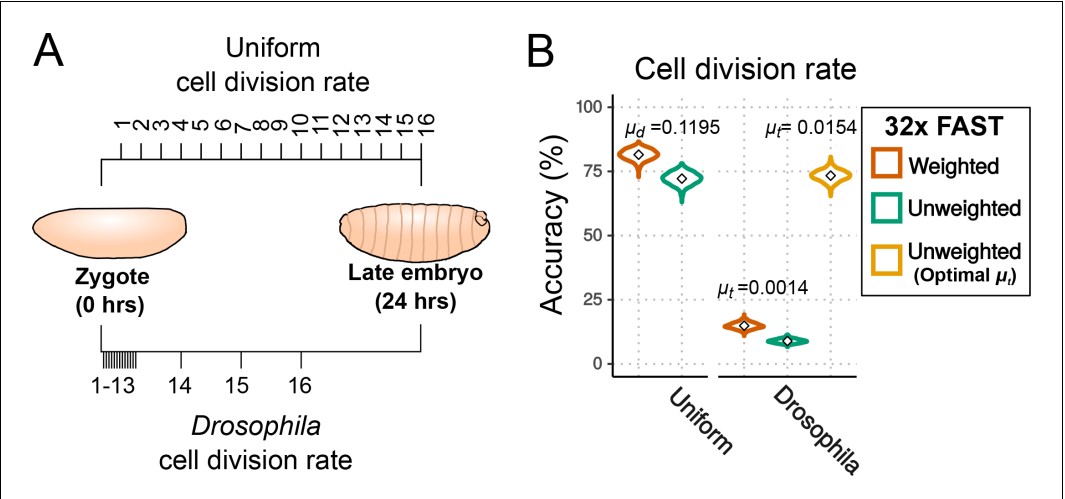

**Figure 5.** Effects of cell division rate on lineage reconstruction. (**A**) Scheme of the comparison between uniform and actual cell division rates in *Drosophila* embryos. (**B**) Accuracy of lineage reconstruction under a uniform cell division rate (left) compared to rates that approximate those actually observed during *Drosophila* development (right) (*Zalokar and Erk, 1976*; *Foe, 1989*), using mutation rates calculated from real experiments for the FAST target ($\mu_t$ = 0.0014), or optimised for accuracy of reconstruction ($\mu_t$ = 0.0154). Violin plots represent the distribution of reconstruction accuracies of 1000 simulations after 16 cell divisions. The accuracy of reconstruction using 32 FAST targets, with or without weighting of mutations, is represented in orange and green respectively. In yellow is the accuracy of 32 targets with an optimal $\mu_t$ (with no weighting).
DOI: https://doi.org/10.7554/eLife.40292.008

The following figure supplement is available for figure 5:

**Figure supplement 1.** Finding the optimal mutation rate for the real rates of cell division in *Drosophila* embryos.
DOI: https://doi.org/10.7554/eLife.40292.009

mutations accumulated in the rapid early cell cycles, and indeed the accuracy fell to just 9% (without using the weighting scheme; see *Figure 5B*).

We hypothesized that the optimal value of $\mu_t$ would be different in this scenario of unequal cell divisions: that a higher $\mu_t$ should improve the accuracy of the reconstructed lineage because it would help to lineage the rapid early cell cycles. To test this hypothesis we performed simulations using different values of $\mu_t$. The results show that the accuracy did indeed improve with increasing rates of mutation (*Figure 5—figure supplement 1*), with a maximum accuracy of 73% at $\mu_t$ = 0.0154 (11 times higher than the optimal rate for embryos with a uniform rate of cell division; *Figure 5B*). These results suggest that higher mutation rates are needed for high lineaging accuracy when the rates of cell division are uneven.

## Modelling the effects of target dropouts

Given the number of targets needed to reconstruct a cell lineage accurately (*Figure 2D*), lineage recorders must include arrays of tens or hundreds of targets. CRISPR activity affecting multiple targets simultaneously, in the same cell, can result in deletions of the DNA between these targets (see *Figure 6*). Such deletions could remove multiple targets, hampering accurate cell lineage reconstruction. We modelled the potential impact of these 'dropouts' on the accuracy of lineage reconstruction, by conducting simulations (with uniform cell divisions, $\mu_d$ = 0.1195 and $m$ = 32, as before) under a scenario in which every time two or more targets were mutated in a given cell at a given cell cycle, we removed all the targets located between them (see Materials and methods). We find that dropouts have a major impact on the accuracy of lineage reconstruction (*Figure 6B*); after 16 cell divisions, the accuracy dropped from 72% to 23% (or from 82% to 26% with weighted mutational outcomes).

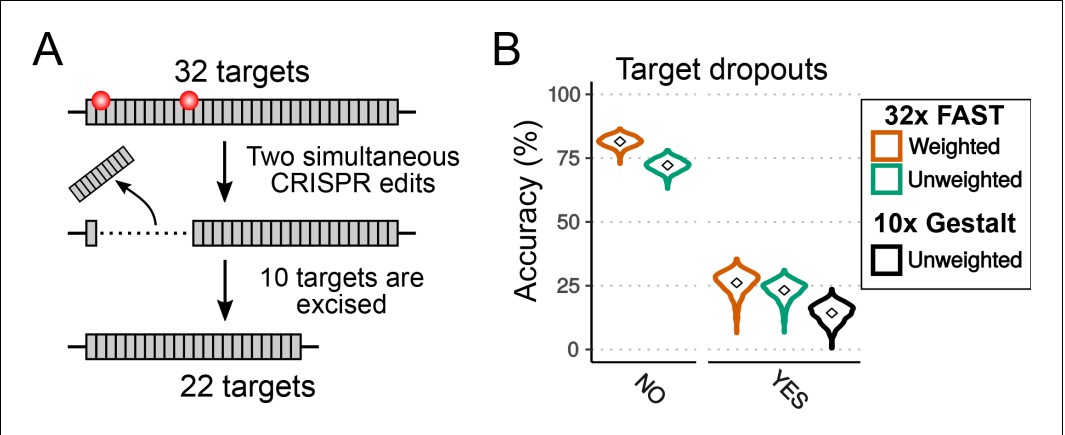

**Figure 6.** Effects of dropouts on lineage reconstruction. (**A**) Schematic representing how targets can be dropped out by simultaneous CRISPR edits. (**B**) Accuracy of lineage reconstruction without dropouts (left) or with the presence of dropouts (right), using a $\mu_d$ = 0.1195. Violin plots represent the distribution of reconstruction accuracies of 1000 simulations after 16 cell divisions. The accuracy of reconstruction using 32 FAST targets, with or without weighting of mutations, is represented in orange and green respectively. In black is the accuracy when simulating the 10 targets of the Gestalt v7 construct (with no weighting; see *Figure 6—figure supplement 1*). DOI: https://doi.org/10.7554/eLife.40292.010

The following figure supplement is available for figure 6:

**Figure supplement 1.** Simulating the mutational outcomes of the GESTALT v7 recorder.
DOI: https://doi.org/10.7554/eLife.40292.011

## Optimising cell lineage reconstruction for *in situ* sequencing with 2, 4 or 16 character states

Besides the biological constraints that influence our ability to reconstruct the cell lineage based on CRISPR recorders (mutation rates, diversity of CRISPR mutations, rates of cell division, target dropouts), there are technical constraints that currently limit our ability to read the information contained in these recorders. Thus far, our simulations have assumed that we can reliably read up to nine nucleotides surrounding each target site over tens of targets, from individual cells. This can be achieved in dissociated single cells using modern high-throughput sequencing technologies (*Spanjaard et al., 2018*; *Alemany et al., 2018*; *Raj et al., 2018*).

Ideally, CRISPR-based lineage recorders could also be used in combination with spatially resolved sequencing (*in situ* sequencing), so that lineage information of single cells could be recorded together with their exact position in the developed embryo. Achieving accurate sequencing of multiple nucleotides in tens of targets in cells *in situ* is currently impractical, however, less ambitious *in situ* approaches have been proposed. The MEMOIR approach (*Frieda et al., 2017*) has addressed this by employing single molecule *in situ* hybridization to distinguish mutated from unmutated targets.

In MEMOIR, only two character states can be detected per target ('scratchpad'), mutated versus unmutated. Moreover, successive rounds of *in situ* hybridization are needed to interrogate many distinct targets, which places a constraint on the number of targets that can be read. (*Frieda et al., 2017*) have shown that three targets can be read per hybridization and up to 9 rounds of hybridization are feasible (*Frieda et al., 2017*); thus, reading two character states per target over ∼30 targets seems to be achievable by the MEMOIR approach.

We carried out simulations to test how MEMOIR would perform using 32 targets, a mutation rate $\mu_d$ = 0.1195 and a readout of 2 character states ('mutated' or 'unmutated'). We find that the accuracy is only 4%. Even with an optimal mutation rate resulting in 50% target saturation (*Frieda et al., 2017*) the accuracy of lineage reconstruction would be only ∼15% (data not shown).

In the future, *in situ* sequencing methods could be developed to interrogate the sequence of each target. These methods would be subject to different technical constraints than MEMOIR. Thus far, *in situ* sequencing efforts have mostly been based on sequencing by ligation and used the SOLiD

sequencing technology (*Lee et al., 2014*; *Ke et al., 2013*), which uses consecutive ligations of fluorescent oligonucleotides to interrogate pairs of dinucleotides in the target sequence sequentially (*Valouev et al., 2008*)). The SOLiD colour code is degenerate, as four colours are used to represent all 16 possible DNA dinucleotides.

As a first step we wanted to explore the SOLiD parameter space extensively, to determine how the number of targets ($m$) and mutation rates ($\mu_d$) affect the accuracy of lineage reconstruction when reading each target with one SOLiD ligation/detection cycle (only four character states) (*Figure 7A*). For this, we performed 10 simulations over a range of values for $\mu_d$ (from 0.01 to 0.3 mutations per cell division) and $m$ (from 10 to 300 targets). In these simulations we assumed that the four possible mutated states were equiprobable and used the complete inferred tree to estimate the accuracy. Our results show (*Figure 7C*) that the optimal mutation rate for lineage reconstruction by this approach lies between 0.05 and 0.12 mutations per cell division, and that is possible to get up to 99% accuracy with 260 targets or more. Using alternative measures of reconstruction accuracy (see below) gives us a very similar estimate of optimal mutation rate (*Figure 7C*)

In SOLiD sequencing, the number of ligation/detection cycles that can be performed is limited by photodamage of the target amplicons and by the time required to perform this type of sequencing (10 days for 30 ligation cycles; *Lee et al., 2015*). The practical upper limit on the number of SOLiD cycles that can be performed is therefore currently in the order of 30–60 cycles. Given these constraints, it is important to optimise the sequencing strategy so as to maximise the amount of sequence information obtained for a given number of SOLiD sequencing cycles. We can ask, for example, whether it would be preferable to perform a single ligation/detection cycle on 64 targets rather than two ligation/detection cycles on 32 targets. Given the experimentally measured spectrum of CRISPR-induced mutations on the targets, we can also determine which nucleotides of the target we should interrogate in order to extract the most information.

We determined that positions 6–7 bp 5' of the PAM sequence yield the most equiprobable colour frequencies for the FAST target (*Figure 7A*), minimising homoplasy in the observed character states (see *Figure 5*). The frequency of each mutated state was determined by the real frequency of mutations observed (see above) and the overall frequency of mutation was set to $\mu_d = 0.1195$ per cell division. We note that the unmutated state (red) is indistinguishable from one of the four mutated states. With 4-character states, homoplasy will arise frequently from convergent appearance of the same colour (even arising from different mutated states) in independent cells. Our results show that, with a single SOLiD read, using a recorder with 32 targets, the mean accuracy of reconstructed cell lineages is 45% (*Figure 7B*).

Clearly, increasing the number of targets will improve performance, but we wanted to know whether it would be better instead to read double the number of nucleotides per target, which represents the same sequencing effort. We found that the reconstruction accuracy obtained by performing 1 SOLiD sequencing cycle on 64 FAST targets is higher (69%) than performing 2 SOLiD cycles on 32 FAST targets (65%) (*Figure 7B*). For the second SOLiD cycle we used the positions 11–12 bp 5' from the PAM sequence, as, in SOLiD, the sequentially interrogated dinucleotide pairs are typically separated by five nucleotides (*Lee et al., 2015*).

## Assessing the accuracy of an existing recorder

Recently, a number of lineaging approaches using CRISPR recorders have been tested in the nematode *Caenorhabditis elegans* and the zebrafish *Danio rerio*, as well as in cultured human cells (*McKenna et al., 2016*; *Schmidt et al., 2017*; *Frieda et al., 2017*). We have used our simulation approach to assess the accuracy of GESTALT, one of the first and most ambitious approaches, which aimed to reconstruct the cell lineage of the tens of thousands of cells of the zebrafish embryo (*McKenna et al., 2016*). It is important to note that in GESTALT the cell lineage is reconstructed at a coarse-grained level, with clones (instead of cells) as nodes in the tree, whereas our measure of success assesses the ability to reconstruct the complete cell lineage at a single-cell level.

GESTALT uses arrays of 10 different CRISPR targets, mutated by injecting fertilised eggs with 10 corresponding sgRNAs and Cas9. The mutated targets are then sequenced at different developmental stages. We based our simulations on the mutational outcomes derived from the GESTALT recorder v7 at 30 hr post-fertilisation (downloaded from the Dryad repository). At this stage the zebrafish embryo consists of approximately 25,000 cells, resulting from ~15 rounds of cell division.

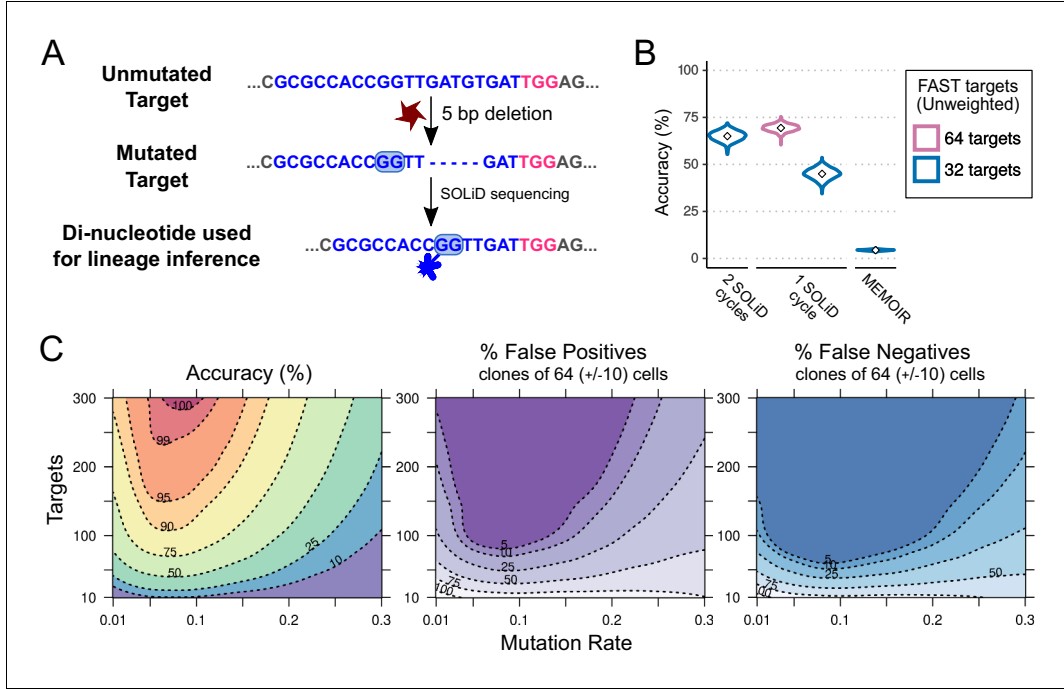

**Figure 7.** Combining CRISPR lineaging with *in situ* sequencing. (**A**) The most common mutational outcome of the FAST target is shown. The coloured box highlights the most informative dinucleotide position to read by SOLiD sequencing (6–7 bp from PAM after CRISPR cleavage) and its SOLiD colour code (see *Figure 7—figure supplement 1*). Sequence colours as in *Figure 4*. (**B**) Accuracy of lineage reconstruction after sequencing with 2 SOLiD sequencing reads (left), 1 SOLiD read (center) and as in MEMOIR (right) using a $\mu_d$ = 0.1195. In blue and pink are the accuracy of a construct with 32 and 64 FAST targets, respectively. (**C**) Accuracy of lineage reconstruction using all cells after 16 cell divisions (N = 65,536) and *in situ* SOLiD sequencing, for different mutation rates and numbers of CRISPR targets, using a $\mu_d$ = 0.1195 and assuming equiprobable colour frequencies after 1 SOLiD read; accuracy (left), false positives (center) and false negatives (right). Dashed lines represent different accuracy thresholds (levelplot) after a LOESS regression. For each parameter combination, we used the mean accuracy of 10 simulations after 16 cell divisions. We found the global accuracy to be similar when subsampling 1,000 cells (*Figure 7—figure supplement 2*).

DOI: https://doi.org/10.7554/eLife.40292.012

The following figure supplements are available for figure 7:

**Figure supplement 1.** Distribution of SOLiD sequencing outcomes on the FAST target, to identify the most informative sites.
DOI: https://doi.org/10.7554/eLife.40292.013

**Figure supplement 2.** Accuracy of lineage reconstruction using a single-read of SOLiD sequencing.
DOI: https://doi.org/10.7554/eLife.40292.014

---

In our simulations, we assumed a constant mutation rate (per cell division), which, as we have shown, will probably overestimate of the accuracy of the inferred lineage. For each of the 10 CRISPR targets, we estimated the mutation rate ($\mu_d$) necessary to obtain the fraction of mutated targets observed after 15 cell divisions (*Figure 6—figure supplement 1*). The estimated mutation rate ranges from ~0.01 (for target 10) to ~0.23 (for target 7) per cell division.

The v7 GESTALT construct shows a high incidence of target dropouts which were modelled as previously described. The mutational process was modelled as a gamma distribution of 60 possible mutated states (*Figure 6—figure supplement 1*), with frequencies closely approximating the observed distribution of mutations reported in the GESTALT publication (see Materials and methods for detail). We compared the number of different alleles (i.e., unique combinations of mutated targets) obtained in the simulated and the experimental results; the mutational complexity used in our simulations generated a number of alleles that closely approximates the experimentally observed number (see Materials and methods for more details).

We performed 1000 simulations and inferred the cell lineage of 1000 randomly sampled cells from each simulation. We find that the mean accuracy of the GESTALT approach is just 14% after 16 cell divisions (*Figure 6B*). This means that this implementation of GESTALT is not suited for reconstructing a complete, accurate cell lineage.

## Measuring accuracy at different depths of the tree

To quantify the decay in number of targets available for mutation, we measured the accuracy (percentage of correct splits) when reconstructing the relationships between groups of four granddaughter cells derived from a single grandmother, sampled after different numbers of cell division (quartet analysis). The only targets that vary amongst the daughters, and hence inform the tree reconstruction, are those mutated in the previous two cell divisions. This analysis allows us to visualise (*Figure 8*) the effect of the decline in available targets as they are mutated, but excludes the effects of subsequent homoplasy or the later loss of targets through dropouts.

To see how this decay in information interacts with other factors, such as dropouts and homoplasy, we applied our Robinson-Foulds related accuracy measure to branches of increasing size (*Figure 9*). When comparing with the quartet analysis, in the simulation with FAST targets and no dropouts, we can see a similar decline in accuracy in later branches of the tree. In simulations with dropouts (with FAST targets and GESTALT) the earlier branches are, in contrast, less well reconstructed than the later ones, presumably due to the loss of informative mutations through dropouts.

As an additional measure of tree accuracy, which could provide a more intuitive estimate of usefulness when thinking about the clonal composition of tissues, we estimated the proportion of false positive (FP) and false negative (FN) assignments of cells to clones in the reconstructed cell lineage (*Figure 9* and *Figure 9—figure supplement 1*). False positives were defined as the proportion of cells that are erroneously assigned to a given cell clone. Conversely, false negatives were defined as the proportion of cells that are not assigned to a cell clone to which they belong. As before, our measurements of false positives and false negatives were performed on trees subsampled from the full simulated lineage and analysed monophyletic groups of cells ('clones') of variable size, as described in the Materials and methods section.

Unsurprisingly, the FP and FN measures closely reflect the topology based accuracy measure. Our results show, for example, that the clones of 33–64 cells reconstructed in simulations based on our FAST construct without dropouts have a high accuracy of 90% and correspondingly low errors, with less than 5% of both FP and FN.

## Discussion

The use of CRISPR-induced somatic mutations is emerging as an attractive approach for reconstructing complex cell lineages. A variety of CRISPR-based lineage recorders has been developed to test this approach (*McKenna et al., 2016*; *Frieda et al., 2017*; *Kalhor et al., 2018*; *Perli et al., 2016*; *Alemany et al., 2018*; *Schmidt et al., 2017*; *Raj et al., 2018*; *Attardi et al., 2018*; *Spanjaard et al., 2018*; *Junker et al., 2016*). If the results of these methods are to be useful for gaining biological insights, however, it is essential that the inferred lineage trees are sufficiently reliable, that is that they accurately reconstruct the real cell lineages of the organism. The required accuracy of a lineage will depend on the application; for example, accurate trees will be necessary to detect stereotypic divisions and cell fates such as those found in *Drosophila* sensory organ precursor and CNS neuroblast lineages, but less accurate trees may be sufficient to detect biases/trends reflecting major lineage commitments. The potential accuracy of the trees inferred using these methods has not yet been established, however.

We have used simulations of the process of cell division and the accumulation of mutations across a lineage tree covering tens of thousands of cells, to examine the effects of different factors on the accuracy of a reconstructed tree. Our simulations allowed us to look at the influence of different rates of mutation on the CRISPR targets, of different designs of lineage recorders and of how mutations could be read experimentally. We have also investigated the effects of irregular cell divisions, target deletions following simultaneous double-stranded cuts and the variable mutational outcomes of the CRISPR process itself.

Unsurprisingly, the accuracy of lineage reconstruction largely rests on the quantity and quality of lineage information carried by the recorders, which is influenced by several factors. Although it is

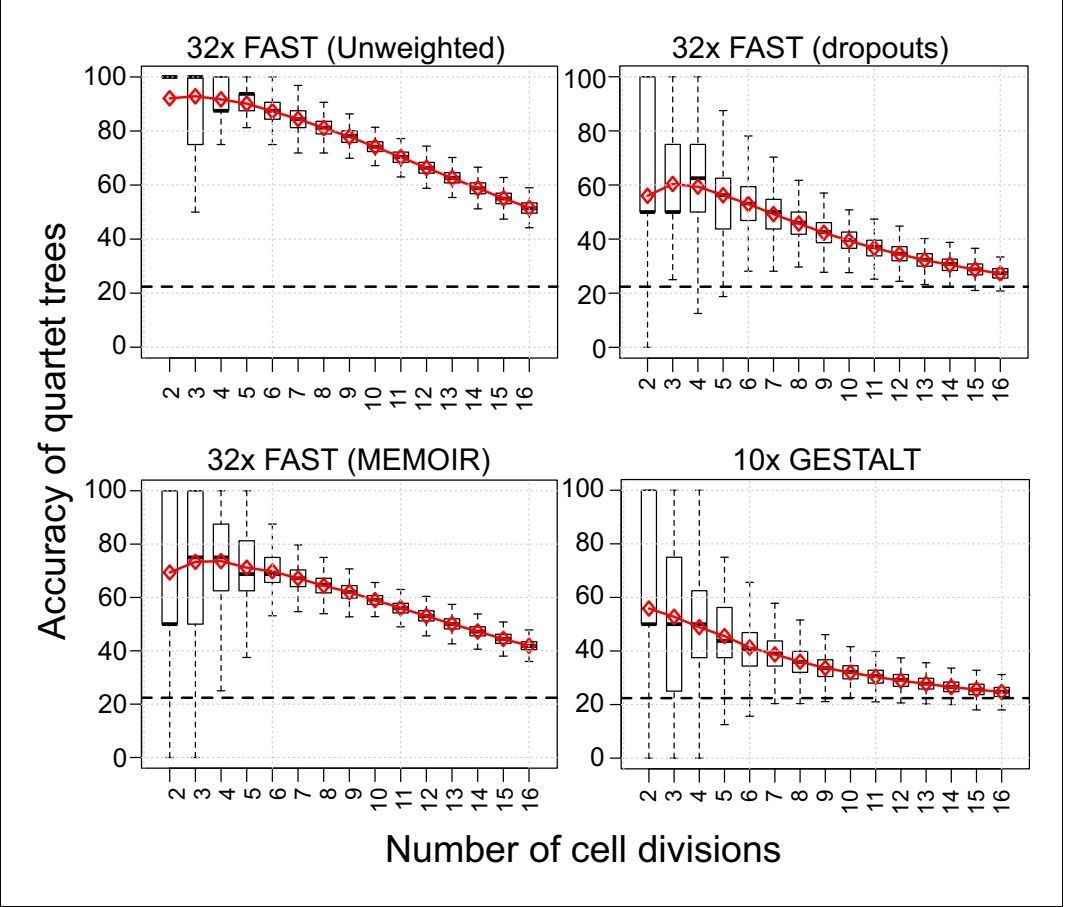

**Figure 8.** Quartet analysis. Boxplots showing the percentage of correct splits in quartet trees measured at cell divisions 2 to 16. Red diamonds represent the average of up to 250 randomly sampled groups of quartet trees from the different simulations. The black dashed line indicates the average percentage of correct splits observed in 100,000 random quartet trees.

DOI: https://doi.org/10.7554/eLife.40292.015

obvious that the accuracy of the lineage tree depends on the number of CRISPR targets in the recorder, our results serve to place strict upper limits on the level of accuracy that we can expect from CRISPR recorders. Under ideal conditions (optimized mutation rates, uniform cell divisions, fully sequenced targets), 30 targets are sufficient to reach an overall tree accuracy of ~70% for a lineage of ~65,000 cells; 100 targets would yield trees that have an accuracy above 90% (*Figure 2D*). If we were only able to take a single 4-colour SOLiD read per target, more than 200 targets would be required to get a highly accurate (>95%) tree (*Figure 7C*).

A second important requirement is to match the mutation rate to the rate of cell division; mutation rates that are too low will leave many cell divisions unmarked, while mutations that accumulate too rapidly will quickly saturate the targets and leave very few available to record later cell divisions. The range of mutation rates that can produce accurate lineage reconstructions fortunately proves to be quite broad for a given tree size; 0.05 to 0.25 mutations per cell division can yield reasonably high levels of accuracy for trees of ~65,000 cells, if the division rates are relatively even (*Figure 2C, D*). This flexibility will be beneficial in cases where the cell division rates are poorly characterised. Alongside the number of targets, mutation rates are an attribute of the experiment that can be adjusted. Rates can potentially be increased by increasing the expression levels of the CRISPR effectors, or decreased by introducing mismatches between the sgRNA and the CRISPR targets, as we have shown experimentally (*Figure 3C*).

The information carried by CRISPR recorders is also influenced by the diversity of the experimentally observed mutations accumulating at each CRISPR target. Our observations of CRISPR mutations

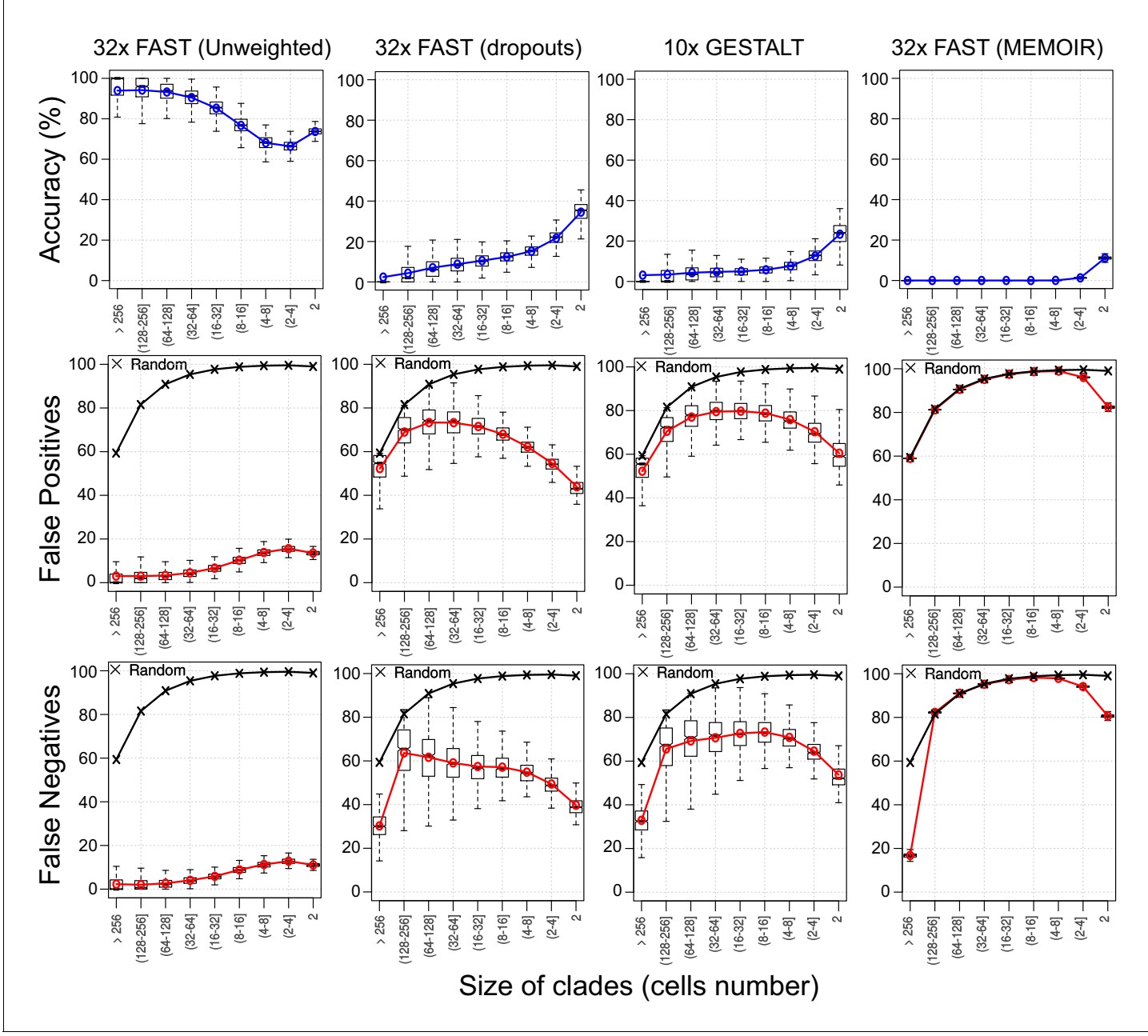

**Figure 9.** Assessing reconstruction accuracy at different tree depths. Three different measures of reconstruction accuracy (rows) are shown for four set of simulations (columns). The x-axis in each plot indicates the size of the clade of cells that is being assessed. In the False Positives and Negatives plots, black crosses show the average values of 1000 random trees.

DOI: https://doi.org/10.7554/eLife.40292.016
The following figure supplement is available for figure 9:

**Figure supplement 1.** Method to estimate false positives and false negatives.
DOI: https://doi.org/10.7554/eLife.40292.017

in *Drosophila* show that these are biased towards a small number of frequently observed outcomes. Simulations show how targets that accumulate a broad set of more equiprobable mutations generate more reliable trees (*Figure 2C*). If, as expected, the diversity of mutations and their relative frequencies vary depending on the target sequence and its local environment (*van Overbeek et al.,*

*2016*; *Vu et al., 2017*; *Allen et al., 2018*; *Chen et al., 2018*; *Shen et al., 2018*) sampling different targets to approach this optimum would be worthwhile.

Some factors affecting tree reconstruction accuracy are outside of experimental control, but simulating their effects can nevertheless show which responses can successfully mitigate them. We have shown, for example, that uneven rates of cell division across the tree require faster mutation rates and/or larger numbers of targets, to provide sufficient coverage during the fastest divisions. In an extreme case, such as the *Drosophila* embryonic lineage where 13 of the 16 cell divisions take place at a very high rate (1 cell division every ~10 min), the optimum mutation rate proves to be >10 times higher than in an equivalent tree with uniform division rates. Even with this optimised mutation rate, the potential accuracy of lineage reconstruction with a given number of targets is much lower (*Figure 5A*).

Besides the intrinsic limitations imposed by CRISPR mutagenesis, the information that we can obtain from each CRISPR target is further constrained by our ability to read and to discriminate between the mutational outcomes. As an obvious goal would be to sequence the mutated targets in individual cells *in situ*, we have explored the specific case of obtaining a single 4-colour SOLiD sequencing read per target. It is encouraging to find that accurate lineage reconstruction is still possible given a sufficient number of targets (*Figure 7C*).

Finally, we show the degree to which the accuracy of lineage reconstruction is sensitive to loss of information caused by the loss of targets through deletion resulting from simultaneous cleavage at two sites (*Figure 6B*). While we have used the most pessimistic estimate of the frequency of dropouts - assuming that every pair of targets cleaved in the same cell would lead to a deletion of the intervening targets in the array - data from GESTALT suggest that target dropouts are frequent when mutation rates are high (*McKenna et al., 2016*). The strong deleterious effect of dropouts that we observe in simulations highlights the need to address this issue. The problem of dropouts could be reduced by opting for the lower end of the optimal range of mutation rates; or eliminated by targeting separate loci in the genome rather than arrays of targets.

Available implementations of CRISPR type recorders are based on different conceptual designs: barcoded arrays recording point mutations (*McKenna et al., 2016*; *Raj et al., 2018*), 'collapsing' arrays (*Frieda et al., 2017*), targets distributed in different genomic locations (*Junker et al., 2016*; *Schmidt et al., 2017*; *Attardi et al., 2018*; *Alemany et al., 2018*; *Spanjaard et al., 2018*) and mutations induced by self-targeting guide RNAs (*Kalhor et al., 2018*; *Perli et al., 2016*). Here we have simulated the first two types of recorders, but we expect that the insights that we have gained on the importance of optimising mutation rates, target numbers and the complexity of character states will apply to all types of recorders.

Our analysis suggests that most of the CRISPR recorders published to date, which rely on at most 10 CRISPR targets (*McKenna et al., 2016*; *Raj et al., 2018*; *Frieda et al., 2017*; *Junker et al., 2016*; *Schmidt et al., 2017*; *Alemany et al., 2018*), yield trees of very low overall accuracy and lineage resolution. While these recorders must, nevertheless, carry lineage information of lower resolution, it is sensible to interpret the results from these recorders in the light of this expected low level of accuracy.

A simulation-guided design of lineage recorders, taking into account the specific parameters of each experimental system, is essential. We hope our study will encourage the general use of simulations of lineage recorders, with the aim of testing their limits, adjusting their design and improving their performance. This approach should stimulate the development of a new generation of CRISPR recorders whose performance will be assessed explicitly and adapted to match the biological questions at hand. Our study predicts that optimised recorders will ultimately allow the reconstruction of accurate cell lineages of complex multicellular organisms at the level of a single cell.

## Materials and methods

### CRISPR recorder and sequencing
Design and synthesis
We designed a DNA construct containing an array of 32 targets of the human Emx1.6 sgRNA (*Hsu et al., 2013*), including the wild-type Emx1.6 target sequence and 31 variants carrying 1 or 2 mismatches and/or an alternative PAM sequence (see *Supplementary file 1—Table1*). To facilitate

synthesis of this construct, between each pair of targets we introduced 80 bp spacers, harbouring unique sequences which would be recognised by specific primers (*Supplementary file 1—Table 3*). We optimised these spacer sequences *in silico* to minimise the presence of repetitive sequences. Unique KpnI and NotI cloning sites were included at either end of the array to help with subsequent cloning steps.

We designed a second plasmid carrying the KpnI and NotI restriction sites and the *Drosophila* U6.2 promoter driving expression of the Emx1.6 sgRNA (*Hsu et al., 2013*) using a standard sgRNA scaffold: GUUUUAGAGCUAGAAAUAGCAAGUUAAAAUAAGGCUAGUCCGUUAUCAAC UUGAAAAAGUGGCACCGAG (*Hsu et al., 2013*; *Port et al., 2014*). This DNA sequence was flanked by two attB sites. The *Drosophila* U6.2 promoter has been shown by previous studies to produce lower levels of CRISPR activity when compared to the U6.1 and U6.3 promoters (*Port et al., 2014*).

Both constructs were synthesised by Biomatik (Ontario, Canada) using standard gene synthesis techniques. The CRISPR target array was excised from the first construct by KpnI-NotI digestion and subcloned into the KpnI and NotI sites of the second plasmid.

## Fly transgenesis, genetics and strains

The construct carrying the CRISPR target array and U6.2::Emx1.6 sgRNA was inserted via recombinase-mediated cassette exchange (*Bateman et al., 2006*) into the second chromosome of *Drosophila melanogaster* (acceptor strain # 27387) using a commercially available service (BestGene Inc, USA).

Homozygous Act-5C-Cas9 females (Bloomington stock # 54590) were crossed with homozygous males carrying the CRISPR target array, set to lay eggs over 30 min intervals in order to obtain synchronised egg collections, and the progeny were collected at different developmental stages (24 hr embryos, third instar larvae, recently hatched adults). As negative controls, to account for sequencing errors, we used adults carrying the CRISPR target array (in heterozygous condition), but lacking the Cas9 transgene.

## DNA extraction, generation of libraries and sequencing

For DNA extraction and sequencing, we pooled approximately 100 embryos, 10 larvae or 20 adults (10 males and 10 females). We extracted genomic DNA by phenol chloroform extraction followed by alcohol precipitation, and generated libraries by PCR using primers harbouring adapter sequences ('fusion PCR') barcoded by condition (see *Supplementary file 1—Table 3*) for sequencing on Ion Torrent Personal Genome Machine (PGM, Life Technologies): as the maximum PGM read length is 400 bp and each target repeat in our construct is 100 bp long, we amplified the repeats in 10 groups of 3 units (amplicons 1–10), plus a group of 2 units (amplicon 11). PCRs were performed using the Phusion High Fidelity Polymerase (New England Biolabs), according to the manufacturer's instructions. Amplicons were run on a gel, excised and purified to eliminate primer dimers. Purified amplicons were then mixed in equimolar amounts and the final pooled mix was sequenced on the PGM sequencer with a 318 v2 chip, as well as a calibration standard to enhance the read quality.

## Filtering of sequencing data

The 7,347,400 reads obtained were de-multiplexed by condition and trimmed to meet quality standards using the Phred software included in the seqtk_trimfq package of the Galaxy software (*Afgan et al., 2016*). We next eliminated sequencing reads that were shorter than 100 bp, lacked the 5' primer sequence, or lacked a target-specific sequence of 11–20 bp downstream of the target (including the PAM) using a custom Python script. In each sequencing read, we used the 9 bp adjacent to the PAM sequence (9mer) to determine whether a target was mutated (the results are shown in *Supplementary file 1—Table 1*).

We quantified sequencing errors (with a custom Python script) by analysing the target sequences of adult flies carrying the CRISPR recorder and the sgRNA but not carrying Act-Cas9 ('untargeted' condition): in these animals we expect any differences from the unmutated state to reflect sequencing errors. In target 16 (FAST target) we found two frequent sequencing errors (single nucleotide deletions) downstream of the target; we decided to include the reads carrying these errors. Targets 17 and 18 did not yield a sufficient number of good quality reads, and targets 13, 21, and 23 showed a high proportion of sequencing errors (*Supplementary file 1—Table 1*).

## Estimating mutation rate and mutational complexity

### FAST target

We estimated the mutation rate of the FAST target based on the proportion of targets that were mutated at the end of embryonic development (86.95%) using a custom Python script based on the geometric cumulative distribution function. The mutation rate of $\mu_d$ = 0.1195 mutations per cell division produces the observed saturation of 86.95% after 16 cell divisions.

We modelled the mutational outcomes of the FAST target based on the mutational outcomes observed at the end of embryonic development (>200 distinct 9mers with frequencies following an exponential curve; see *Figure 4C*). We considered that a mutation would result in a change to one of 59 states with a probability reflecting the observed occurrence of the 59 most frequent real mutations (95% of the total; see *Figure 4C* and *Figure 4—figure supplement 1*) or to a 60th state with a probability of 0.05. For the simulation of SOLiD sequencing we used the real distribution of the dinucleotides observed at positions 6–7 and 11–12 upstream of the PAM. We translated each dinucleotide into one of four different colour states according to the SOLiD protocol (*Figure 7—figure supplement 1*).

### Gestalt

To analyse the accuracy of GESTALT, we used data from the v7 construct at the 30 hr post-fertilisation stage (available at https://datadryad.org/resource/doi:10.5061/dryad.478t9). These consist of six biological replicates. The v7 construct contains 10 different CRISPR targets that were targeted with 10 different sgRNAs.

For each biological replicate we quantified the frequency of mutations and dropouts in each target (*Figure 6—figure supplement 1*) using a custom Perl script. We considered any deletion greater than 26 bp to be a dropout, as this would affect more than one target (each target is 23 bp). For each target, we quantified saturation as the proportion of reads of the target that were mutated. For each target, we estimated the mutation rate per cell division ($\mu_d$) necessary to produce the level of saturation (proportion of mutated targets) observed after 15 cell divisions, assuming that mutations follow a geometric distribution. The estimated mutation rate ranges from ~0.01 (target 10) to ~0.23 (target 7) (*Figure 6—figure supplement 1*).

The mutational complexity varied between targets and replicates, from ~25 to ~200 different mutations per target. In all cases, however, their frequencies followed an exponential curve, with one mutation usually accounting for 20–30% of the total reads and with the majority of the mutations observed only rarely. For each target we modelled the mutational outcome as 60 different mutations with frequencies sampled from a random gamma distribution, with shape parameter $\kappa$ = 0.1 and scale parameter $\theta$ = 2, which approximate the observed distribution (see *Figure 6—figure supplement 1D*).

## Computer simulations

Computer simulations were performed using MATLAB v2017a (Mathworks, 2017) and are available at a Github repository (*Salvador-Martínez, 2018*; copy archived at https://github.com/elifesciences-publications/CRISPR_recorders_sims). CRISPR mutations were simulated following a geometric or a poisson distribution.

### Simulating mutation events using a geometric distribution

To simulate mutations using a geometric distribution, the probability of mutation was the same for all targets per cell division. Given a mutation rate $\mu_d$ (per cell division), the probability that a site remains unmutated after $d$ cell divisions is $(1 - \mu_d)^d$. Thus, we can determine the mutation rate $\mu_d$ from the proportion of targets that are mutated after a given number of cell divisions.

### Simulating mutation events using a poisson distribution

Under the Poisson model, given a mutation rate $\mu_t$ (per minute), the probability that a site remains unmutated after $t$ minutes is: $e^{-(\mu_t t)}$. Thus, we can determine the mutation rate $\mu_t$ from the proportion of targets that are mutated after a given amount of time. The time interval for each cell division was set to approximate the rates of cell division in early *Drosophila* embryos: for the first 13 cell

divisions the interval was set to 10 min, and to 130 min per division for the last three divisions (*Zalokar and Erk, 1976*; *Foe, 1989*).

## Simulation of target dropouts
For the dropouts simulations, if any two targets were hit during a given cell division, all the targets between them were removed. When three or more targets were hit during the same cell division, two were selected randomly and the intervening targets were removed. In subsequent phylogenetic analyses, dropouts were treated as missing data.

## Simulations of GESTALT
We performed 1000 simulations with the estimated $\mu_d$ for each target over 16 cell divisions. We accounted for dropouts as described previously. To test whether our simulations match the experimental results in terms of mutational complexity, we compared the number of 'alleles' (unique combinations of mutated targets) found in the experimental and in the simulated data.

Our simulations encompassed 15 cell divisions, yielding 32,768 cells, which approximates the 30 hpf zebrafish embryo (~25,000 cells). For each simulation, we took 100 random samples of 10,000 cells and counted the number of alleles in each sample. Our simulated samples produced an average of 3022 alleles (s.d. = 877 alleles; see *Figure 6—figure supplement 1*), compared to the 1,000–2,500 alleles found in the experimental data (*McKenna et al., 2016*).

## Analysis of simulated targets
The main outcome of each simulation was a $T$ matrix of size $N \times m$, for $N$ cells and $m$ targets. This matrix is equivalent to a DNA alignment with species as rows and DNA positions as columns. For most simulations, 10 random samples of 1,000 cells were chosen for lineage reconstruction and for assessing the accuracy of the reconstructed cell lineage. A 'root' taxon with unmutated character states was added to the alignment prior to the lineage inference. For some simulations, we inferred cell lineages using all cells after 16 cell divisions ($N = 65,536$) and found that their global accuracy was similar to that when subsampling 1,000 cells (*Figure 7—figure supplement 2*).

For the target dropouts simulations we added to the $T$ matrix an extra character for each distinct dropout of one or more targets that was shared between $\geq 32$ cells (character state '1' if present, '0' if absent). This was done to take advantage of the information coming from shared target dropouts.

## Cell lineage inference
### Reconstructing lineage trees using neighbor joining (PAUP*)
Most cell lineages were inferred using the Neighbor-Joining method (NJ). We used the Neighbor joining algorithm as implemented in the PAUP* software (version 4.0a build 158; (*Swofford, 2017*)). In PAUP*, up to 64 character states can be specified, with the possibility of giving different weights to the occurrence of specific mutations. We used a substitution matrix based on the frequency of each mutation. The matrix has size $s \times s$ for $s$ number of states, where the distance from state $i$ to state $j$ is weighted according to the natural logarithm of the inverse of their frequencies (*Felsenstein, 1981*) with the equation:

$$d(i,j) = \begin{cases} 0, & \text{if } i = j \\ log(\frac{1}{Freq_j}), & \text{if } i = unmutated \\ log(\frac{1}{Freq_i}), & \text{if } j = unmutated \\ log(\frac{1}{Freq_i}) + log(\frac{1}{Freq_j}), & \text{otherwise} \end{cases}$$

where $Freq_i$ and $Freq_j$ are the frequencies of states $i$ and $j$ respectively.

In simulations where we modelled dropouts, an extra character state was assigned to each cell containing a dropout that was shared by $\geq 32$ cells. For these simulations, the distance matrix was applied to the $m$ original targets and for the extra characters the following distance was applied:

$$d(i,j) = \begin{cases} 0, & \text{if } i = j \\ 100, & \text{otherwise} \end{cases}$$

For samples of up to 180 cells we compared the accuracy achieved using Neighbor Joining with

an alternative approach using a parsimony method. We show that, while accuracy is consistently slightly higher using parsimony, the average time taken to reconstruct a tree of only 180 cells is on average ~52 hr compared to less than 1 s using Neighbor Joining (see *Figure 2—figure supplement 1*).

## Reconstructing lineage trees using FastTree

When inferring complete cell lineage trees in the simulations of SOLiD sequencing data (*N* = 65,536 cells), we used an heuristic method that approximates the Maximum Likelihood approach, implemented by the FastTree software (*Price et al., 2010*). FastTree was chosen for its ability to infer trees from large alignments, consisting of tens of thousands of sequences, and for doing so very efficiently.

## Reconstructing lineage trees using maximum parsimony (PAUP*)

The use of parsimony for the cell lineage reconstruction of our simulations was not practical for the thousands of cells/taxa we consider. Nevertheless, to assess the relative performance of NJ and Maximum Parsimony in the context of lineage data, we compared the two methods using trees of variable size of randomly sampled cells, from 10 to 180 (*Figure 2—figure supplement 1*). We calculated the accuracy of lineage reconstruction with NJ and parsimony methods on 10 separate simulations, based on the mutational frequencies of FAST targets, four character states and a mutation rate of 0.1195 over 16 cell divisions. For the parsimony analysis, we used the Camin-Sokal model (i.e., irreversible mutated states) and a substitution matrix based on character states frequencies (as used in GESTALT).

## Tree accuracy estimation

### Robinson-Foulds related measure of accuracy

The accuracy of each cell-lineage reconstruction was determined by calculating a measure related to the Robinson-Foulds distance between the reference and the inferred trees. We counted the percentage of splits (sets of cells separated into two groups by a node in the tree) in the reference tree that were also found in the inferred tree. For this task we used the CompareTree software (CompareTree.pl is available at http://www.microbesonline.org/fasttree/treecmp.html).

### Calculating false positives and false negatives

False positives (FP) and false negatives (FN) were calculated by comparing the reference tree (*R*) with the inferred tree (*I*) using the newick-tools software (*Flouri et al., 2018*) using the –difftree command, which analyses the split differences between two trees. The options –filter_eq and –filter_gt were used to specify the number of cells *n* (or a given range) to be analysed (see the newick-tools reference for details).

False positives were measured by counting the proportion of cells that need to be pruned from a branch of the inferred tree to match a given branch of *n* cells in the reference tree. More formally, the false positives were estimated as follows (see *Figure 9—figure supplement 1*):

1. We extracted the *x* number of subtrees in *R* that contained *n* cells (subtrees $R'_{(1-x)}$)
2. Then for each *R'* subtree we find the subtree from *I* (*I'* subtree) that includes all the cells present in *R'*. The FP is then calculated with the following equation:

$$FP_{(R,I)} = \frac{1}{x}\sum_{i=1}^{x}\frac{[I'_i] - [R'_i]}{[I'_i]} \tag{1}$$

where $[R'_i]$ and $[I'_i]$ are the number of cells in trees $R'_i$ and $I'_i$ respectively.

False negatives were measured by counting the proportion of cells that need to be pruned from a branch of the reference tree to match a given branch of *n* cells in the inferred lineage tree. More formally, the false negatives were estimated as follows (see *Figure 9—figure supplement 1*):

1. We extracted the *x* number of subtrees from *I* that contained *n* cells (subtrees $I'_{(1-x)}$).
2. Then for each *I'* subtree we extracted the subtree from *R* (*R'* subtree) that included all cells from the *I'* tree. The FN is calculated then with the following equation:

$$FN_{(R,I)} = \frac{1}{x}\sum_{i=1}^{x}\frac{[R'_i] - [I'_i]}{[R'_i]} \tag{2}$$

where $[R'_i]$ and $[I'_i]$ are the number of cells in trees $R'_i$ and $I'_i$ respectively.

## Quartet analysis

We measured the accuracy (percentage of correct splits) when reconstructing the relationships between groups of four granddaughter cells derived from a single grandmother, sampled after different numbers of cell division. We considered sets of 4 granddaughter cells (each cell represented as the array of targets with accumulated mutations) derived from their single most recent common ancestor. We reconstructed a tree using NJ, rooting the tree with an outgroup, represented by an array of unmutated targets. We calculated the percentage of splits from the reference tree also present in the inferred tree. In cases when a simulation had fewer than 1,000 cells (divisions 2–9) we considered all possible quartets. When there are more than 1,000 cells (divisions 10–16) we extracted 250 random 4 cell trees (total cells = 1000).

## Acknowledgements

We thank Sandrine Hughes and Benjamin Gillet for expert assistance in sequencing the CRISPR target arrays, Tomas Flouri for helping implement the false positives/negatives analysis, Ziheng Yang for advice on phylogenetic reconstruction, Je Hyuk Lee, Jessica Svedlund and Mats Nilsson for discussions on *in situ* sequencing, Nikos Konstantinides, Pierre Martinez, James Cotterell, Rosa Barrio, Michael Akam, Isaac Salazar-Ciudad and Richard Copley for critical comments on the manuscript, and the High Performance Computing platform at the UCL Department of Computer Science. This research was supported by a grant from the Human Frontier Science Programme (HFSP RGP0002/2016).

## Additional information

### Funding

| Funder | Grant reference number | Author |
| --- | --- | --- |
| Human Frontier Science Program | HFSP RGP0002/2016 | Irepan Salvador-Martínez Marco Grillo Michalis Averof |

The funders had no role in study design, data collection and interpretation, or the decision to submit the work for publication.

### Author contributions

Irepan Salvador-Martínez, Conceptualization, Data curation, Software, Formal analysis, Investigation, Visualization, Methodology, Writing—original draft, Writing—review and editing; Marco Grillo, Conceptualization, Resources, Investigation, Methodology, Writing—original draft, Writing—review and editing; Michalis Averof, Maximilian J Telford, Conceptualization, Supervision, Funding acquisition, Methodology, Writing—original draft, Project administration, Writing—review and editing

### Author ORCIDs

Irepan Salvador-Martínez (iD) http://orcid.org/0000-0002-3188-9784
Marco Grillo (iD) http://orcid.org/0000-0003-2155-0645
Michalis Averof (iD) http://orcid.org/0000-0002-6803-7251
Maximilian J Telford (iD) http://orcid.org/0000-0002-3749-5620

### Decision letter and Author response

Decision letter https://doi.org/10.7554/eLife.40292.023

Author response https://doi.org/10.7554/eLife.40292.024

## Additional files

### Supplementary files

• Supplementary file 1. Supplementary Table 1. Proportion of mutated targets (target saturation) for each of the 32 Emx1.6 target variants, sampled at different developmental stages (embryos, L3 larvae, adults) and in the absence of Cas9 (untargeted). Targets 13, 17, 18, 21 and 23 were not analysed further because there were no good quality reads in the untargeted condition or because the targets showed a high proportion of sequencing errors. Supplementary Table 2. Proportion of mutated targets (target saturation) in the embryo after correcting for sequencing errors and estimated mutation rates per cell division, for the target variants showing the highest mutation rates. Supplementary Table 3. PCR primers used for preparation of the sequencing libraries. Forward primers (F) carry adapter sequences (uppercase), barcodes specific for each condition (underlined, BC1 to BC6), and sequences annealing to the spacers of the repeat construct (lowercase). Reverse primers (R) carry adapters (uppercase) and sequences annealing to the spacers of the repeat construct (lowercase); see *Figure 3B* and Materials and methods.
DOI: https://doi.org/10.7554/eLife.40292.018

• Transparent reporting form
DOI: https://doi.org/10.7554/eLife.40292.019

### Data availability

The sequencing data for the in vivo assessment of mutagenesis rates are available at: doi:10.5061/dryad.qb7r0d3. The scripts used to generate all the simulations used in this work, for the analysis of the sequencing reads and for the analysis of the GESTALT construct are available at the Github repository https://github.com/irepansalvador/CRISPR_recorders_sims (doi: doi.org/10.5281/zenodo.1320964; copy archived at https://github.com/elifesciences-publications/CRISPR_recorders_sims).

The following dataset was generated:

| Author(s) | Year | Dataset title | Dataset URL | Database and Identifier |
|---|---|---|---|---|
| Salvador-Martínez I, Grillo M, Averof M, Telford MJ | 2018 | Sequencing data from 'Is it possible to reconstruct an accurate cell lineage using CRISPR recorders?' | http://dx.doi.org/10.5061/dryad.qb7r0d3 | Dryad Digital Repository, 10.5061/dryad.qb7r0d3 |

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
