## [Decision Letter]

[**Editorial note:** This article has been through an editorial process in which the authors decide how to respond to the issues raised during peer review. The Reviewing Editor's assessment is that all the issues have been addressed.]

Acceptance notification:

Prof Telford and colleagues have greatly improved their thoughtful manuscript exploring the limitations and key experimental/analytical parameters of CRISPR-based recorders for lineage tracing, and appropriately addressed the reviewers' comments. Importantly, this study now includes an expanded assessment of lineage accuracy, false positive and negative reconstruction events, and choice of cell lineage inference method. This is a very useful addition to the field that informs optimized design of next-generation CRISPR recorders.

Decision letter after peer review:

Thank you for submitting your article "Is it possible to reconstruct an accurate cell lineage using CRISPR recorders?" for consideration by *eLife*. Your article has been reviewed by three peer reviewers, one of whom is a guest Reviewing Editor, and the evaluation has been overseen by Aviv Regev as the Senior Editor. The following individuals involved in review of your submission have agreed to reveal their identity: Aaron McKenna and Jay Shendure co-reviewed as Reviewer #3. The other reviewers remain anonymous.

The Reviewing Editor has highlighted the concerns that require revision and/or responses, and we have included the separate reviews below for your consideration. If you have any questions, please do not hesitate to contact us.

As you will see, the reviewers agree this work is a valuable addition to the rapidly growing field of CRISPR recorders and lineage tracing. They also each raise concerns about the results presented and their interpretation. In particular, there was clear consensus about the appropriate definition of and requirements for "accurate" lineage reconstruction, justification of the tree reconstruction algorithms used, and the methodologies used for analysis of editing in *Drosophila* target arrays. Reviewer 2 also has specific suggestions for the title. We hope these points will be straightforward to assess in a revised manuscript.

Separate reviews (please respond to each point):

*Reviewer #1:*

There has been an explosion in CRISPR lineage tracing studies over the past few years. However, assessing the "accuracy" of these reported lineage reconstructions is difficult because the correct lineage is typically unknown. In this study, the authors primarily use computational approaches, along with some experimental data in *Drosophila* embryos, to assess various parameters that affect reconstruction accuracy, such as mutation rate, cell division rate, mutational diversity, and others.

Some thoughts for the authors to consider: From a practical perspective, what is accurate enough? In an experiment where the "real" tree is unknown, how do the investigators determine the appropriate parameters? Are some CRISPR recorder designs fundamentally better than others, and does this study suggest a better approach?

Major comments:

1) Recently, there was a systematic follow up to van Overbeek et al., 2016, by Leopold Parts at the Sanger (Allen BioRxiv 2018), which used a large dataset to provide an indel distribution prediction tool. It would be very interesting to consider this for specific target sequences, and discuss "optimized" spacer sequences for CRISPR recorders that have higher mutational diversity than others.

2) The rationale for assuming that CRISPR recorders generate irreversible target mutations is unclear. For example, there can be a "back rate" where a mutated target that creates a single base indel can still be recognized by Cas9 after a cell division.

3) Why is neighbor-joining chosen over parsimony (and over several other possible algorithms?) This should be elaborated on. Can the authors leverage their insights to improve these reconstruction algorithms to specifically address the challenges of CRISPR recorders?

4) The chosen definition of "accuracy" seems to be problematic because it does not generally discriminate between false positive and false negative reconstruction events. This is briefly considered in Figure 7 but should be expanded.

5) The authors show that setting the correct mutation rate matters. How can mutation rates be matched to the rate of cell division when the rate or interval of cell divisions is unknown/itself variable (e.g. in a tumor)? This would be worth discussing.

*Reviewer #2:*

The manuscript "Is it possible to reconstruct an accurate cell lineage using CRISPR recorders?" uses simulations and some experimental data to conduct a thorough exploration of the parameters governing the accuracy of CRISPR lineage tracing. They define and sweep four key parameters for two types of CRISPR recorders: number of targets, mutation rate, mutational character states, and dropouts. This work is important and valuable to a growing community exploring the potential of CRISPR recorders. It provides a path for future optimization of these type of lineage tracing tools. For the most part, the writing and figures are clear and informative. I have five major concerns listed below.

1) The provocative title in the form of a question is somewhat misleading – is the answer yes? The manuscript doesn't explicitly answer this question. I would suggest a more accurate title, for example "design specifications for more accurate CRISPR recorders."

2) I have two concerns with the term "accuracy."

First concern: The authors should be explicit upfront about their definition of the term accuracy, which includes both false positives and false negatives. I would argue that false positives (wrong branchpoints) are worse than false negatives (i.e. missing branchpoints). Can't an "accurate" tree have missing branchpoints but no wrong branchpoints? The authors briefly describe this distinction, but only at the very end of the Results section and then calculate these subscores of "accuracy" only for the SOLID sequencing approach. I would appreciate a longer discussion of this definition and scoring at the beginning of the results, and FP/FN calculated and reported for all simulations.

Second concern: Generally speaking, how accurate do we need CRISPR recorders to be? The authors set a high bar: complete and accurate lineage tracing of a 65,000 cell tree. Is 4% or 14% (the final "scores" given to MEMOIR and GESTALT) good enough for many scientific applications? It is certainly better than nearly all existing lineage tracing techniques, from which we have learned a great deal about biology. I would appreciate a discussion of why 100% complete and accurate trees are such an important goal, and what we can still learn from less accurate trees.

3) I am concerned about three library prep or filtering steps in the sequencing of the *Drosophila* target site array that are not clearly explained in the text, which may reduce the number of character states used in all following simulations:

a) To call character states – why only use 9bp flanking the target site instead of simply aligning each read to unedited sites? I would assume that this would eliminate some character states.

b) If I understand Figure 4A, the authors use a primer that sits directly on the PAM -mutational outcomes that disrupt any bases within the PAM presumably would not be captured by the PCR and sequencing. Can the authors discuss the impact of this?

c) Why merge 140 rare character states into a shared state (state 60)? This would obviously lead to tree errors. Why not treat these just as all other character states? The authors write that this was "for convenience," but I'm not sure why this is convenient and it seems to be a potential source of false positives.

I believe other papers have shown a higher number of potential character states, and I'm concerned these steps listed above may impact the simulations.

4) The authors, as they admit, use "the most pessimistic estimate" of the frequency of dropouts. It is entirely possible to have two cuts within a cell cycle result in two edits instead of a dropout. Have the authors considered explicitly using their own experimental data (even with library prep caveats), or the GESTALT data, to simulate dropout rates? Since this assumption dramatically impacts accuracy, it would seem important to be careful about how to model dropouts.

5) Choice of tree reconstruction algorithm matters. In their own simulations, Parsimony appears more successful that Neighbor Joining (Figure 2—figure supplement 1). However, this was never stated explicitly in the text, and there is no discussion of algorithm selection and its impact on accuracy. While NJ was selected for obvious reasons (speed), the authors should provide a clear discussion of other options and their impact on tree accuracy.

Minor Comments:

1) GESTALT cell culture approach also used a similar strategy of an array of off-targets and 1 guide RNA, and should be cited when discussing the fly array design, along with the caveats associated with this approach (poor editing at many off targets).

2) The MEMOIR paper extensively considers accuracy of tree reconstruction, including comparisons to reference trees. These data and discussions should be mentioned and referenced in this paper.

*Reviewer #3:*

In this manuscript, Salvador-Martínez and Grillo et al. present a simulation study of newly developed CRISPR lineage tracing technologies. The authors do a good job of setting up the problem, explaining their choices of various parameters and assumptions, and adding experimental data in *Drosophila* to reinforce these choices. This work will be a valuable addition to a quickly advancing field, particularly as a reality check on the extent of organismal engineering that will likely be required to achieve near-complete, accurate trees by this class of methods. Our first major comment is about the tone of the paper, while our our additional major comments primarily relate to alternative measures that should be evaluated that strike more of a balance between strict accuracy and the general conservation of tree topology. Additionally, some basic flaws in the experimental design and analysis of editing outcomes in *Drosophila* should be addressed before publication.

1) A first major comment is that the measure of accuracy used throughout the paper is very conservative, and more generally the tone that is struck in many parts of the paper is (we feel) overly conservative. Although strict accuracy and completeness are of course goals worth shooting for, they are not prerequisites these kinds of experiments to achieve biological insights. For example, lineage relationships between cell types might well be accurately inferred from the general topology of a tree that contained inaccuracies or uncertainty near its tips. For any new technology, proof-of-concept studies are just that -- proof-of-concept, and it's always been clear that significant additional engineering would be (and still is) required to maximize the value of these methods. This point does not detract from the value of the simulations presented in this paper. A more optimistic take on the same results is that it is possible to reconstruct large trees with reasonably high accuracy (great!), but it will require the introduction of at least 50 targets (and ideally several hundred targets), tuning of the mutation rate (although the broad plateau presented in Figure 2D is rather encouraging), and careful consideration of variable cell division rates. These conclusions and other analyses presented in the paper provide important guidance for the field (and a reality check against short term thinking), but the paper often slips into a negative tone that in our view is inconsistent with the results themselves (e.g. the fact that conditions are identified where reconstruction with 99% accuracy is achieved; subsection “Optimising cell lineage reconstruction for in situ sequencing with 2, 4 or 16 character states” paragraph six). We urge the authors to: (a) make it clearer, from the beginning of the paper, the extent to which accuracy as defined here is a highly conservative definition, relative to what might be required to achieve biological insights from trees reconstructed from GESTALT or related methods. (b) strike a more balanced tone, with less emphasis on what is not possible using the systems as reported in their proof-of-concept implementations, and more emphasis on the path forward, i.e. the extent to which further engineering (more targets, reducing inter-site deletion, tuning of mutation rate, etc.) is required to get the most out of these methods.

2) The simulation of CRISPR-based lineage tracing technologies in the first section (Figure 2) focuses on accuracy with Robinson-Foulds, but should include characterization with the clonal reconstruction measurements (FP / FN analysis). These measurements are used later in the MEMOIR simulation section, and many people will be interested in this use of lineage tracing technologies for this purpose. A mention or comparison to other distance metrics might be more appropriate, see https://www.ncbi.nlm.nih.gov/pubmed/21383415 or the review https://www.ncbi.nlm.nih.gov/pubmed/25378436

3) You show higher accuracy with maximum parsimony approaches, but use neighbor-joining throughout the paper. A strong justification of this choice is important, as this appears to be a strong bias against the methods you're evaluating.

4) Robinson-Foulds is a distance metric, and here you've normalized it to an accuracy over [0-100%], which is not detailed anywhere in the paper. Details in the Materials and methods section would improve the clarity of the paper.

5) Counting 9mers is a bit liberal for determining editing outcomes from the FAST target sequencing. Given the known double-stranded break location and repair outcomes, most mutational outcomes should be centered at, or overlap the cutsite. By including all of the 9 proximal bases, the captured editing diversity will include sequencing errors and PCR errors. This shouldn't affect the overall profile of editing outcomes, but will increase the number of mutations, and should be mentioned. This is made apparent by the elevated mutation rates in the untargered column of Supplementary Table 1 in Supplementary file 1, as some of the more active off-targets have changes in bases believed to strongly obstruct Cas9 binding (bases that are very close the PAM sequence).

6) Also the location of the primer (Figure 4) precludes find deletions that extend downstream into the PAM sequence. This eliminates the detection of mutations that extend 3' of the cutsite, and will deflate the diversity of editing you see. This is a major bias, and could also affect the FAST off-target results.

7) It's been shown that very large deletion frequency decreases with the distance between two cutsites (https://www.ncbi.nlm.nih.gov/pubmed/24907273). it would be worth including in the dropout simulations, or at least mentioning an alternative model with decreasing dropout efficiency at larger distances.

8) It would be good to characterize the accuracy of genotype collapsed trees (https://www.ncbi.nlm.nih.gov/pubmed/29474671) which aim to reduce the number of false branch points (branch points introduced by the tree bifurcation requirement, not mutations).

9) In the SOLiD simulation section (for Figure 7) it's unclear what sequence would be used for the primer, and how often that binding sequence would be obstructed by deletions in its target sequence. Some more details here would be helpful.

Minor Comments:

Fourth sentence of the Abstract should probably say approaches (you profile at least two).

Abstract: not all terminal branches are fully differentiated cells.

Introduction paragraph two: there's probably a better phase than “simple cases” (it was quite the effort, which you mention further on).

Introduction paragraph eight is a bit strong. Certainly simulations will inform future synthetic recording systems, but the assumptions and simplifications of simulation might prevent finding the optimal solution without more true biological validation.

Subsection “Optimising cell lineage reconstruction for in situ sequencing with 2, 4 or 16 character states” paragraph nine: shouldn't "double the number of reads per target" be "double the read length" or "double the number of cycles"?

Really beautiful figures all through the supplement, but the most minor change, add " + theme_classic()" or similar to the R command for Figure 2—figure supplement 1.

Additional details are needed in the experimental section about the PCR reaction.

---

## [Author Response]

As you will see, the reviewers agree this work is a valuable addition to the rapidly growing field of CRISPR recorders and lineage tracing. They also each raise concerns about the results presented and their interpretation. In particular, there was clear consensus about the appropriate definition of and requirements for "accurate" lineage reconstruction, justification of the tree reconstruction algorithms used, and the methodologies used for analysis of editing in Drosophila target arrays. Reviewer 2 also has specific suggestions for the title. We hope these points will be straightforward to assess in a revised manuscript.

We thank the reviewers for their constructive comments. We have revised the manuscript to address and to clarify the points that have been raised, paying particular attention to estimating and interpreting accuracy, and to clarifying our methodology.

Important messages of our paper are that: there will always be a minimal level of accuracy that must be achieved if reconstructed lineage relationships are to be interpreted confidently; that this will be problem dependent; and that this requirement needs to be considered when designing a lineaging experiment.

Having a measure of lineage reconstruction accuracy is the essential basis for comparisons of different designs of recorders and for anticipating the effects of different biological factors.

All reviewers commented on the meaning of the accuracy metrics, asking what level of accuracy would be required for this lineaging approach to be useful. We cannot provide a uniform answer to this question because the level of accuracy required will depend on the questions being asked and the biological conclusions drawn from the cell lineage. Researchers will need to assess the level of lineage accuracy needed to answer a given question, on a case-by-case basis.

To extend and to clarify our measurements of lineage accuracy, in the revised manuscript we include the following new results:

1) In addition to our global measure of accuracy, we provide estimates of lineage accuracy at different levels of the lineage tree. This gives an indication of accuracy for large (early) clones versus smaller (later) clones.

2) We developed a new measure of reconstruction accuracy that we are able to apply at all levels of the lineage tree to supplement our global measure of tree accuracy. This measure can show how the information content within the targets decays during development, as the targets become saturated.

3) We have extended the estimates of False positives and False Negatives (a measure the reviewers agree is useful for understanding the impact of accuracy) to all our major simulations, and we provide estimates of false positives and false negatives for different clone sizes.

Separate reviews (please respond to each point):

Reviewer #1:

There has been an explosion in CRISPR lineage tracing studies over the past few years. However, assessing the "accuracy" of these reported lineage reconstructions is difficult because the correct lineage is typically unknown. In this study, the authors primarily use computational approaches, along with some experimental data in Drosophila embryos, to assess various parameters that affect reconstruction accuracy, such as mutation rate, cell division rate, mutational diversity, and others.Some thoughts for the authors to consider: From a practical perspective, what is accurate enough?

We have as our ultimate goal an accurate lineage tree for an entire embryo but recognise that less accurate/resolved trees can also be useful, depending on the biological questions being asked.

We clarify this in the revised manuscript, as follows:

1) Already in the Introduction we state "Of course the required accuracy will depend on the intended use of the lineage".

2) In the Discussion we have added:”The required accuracy of a lineage will depend on the application; for example, accurate trees will be necessary to detect stereotypic divisions and cell fates such as those found in *Drosophila* sensory organ precursor and CNS neuroblast lineages, but less accurate trees may be sufficient to detect biases/trends reflecting major lineage commitments.”

In an experiment where the "real" tree is unknown, how do the investigators determine the appropriate parameters?

In a sense, this is what we are doing, as the embryonic lineage tree of *Drosophila* is unknown (in fact the lineage is not stereotypic, it varies in every individual). We are getting as much information as possible from experimental data and knowledge of fly biology, and include these parameters (mutation rate, mutational diversity, cell division rates) in our simulations to model the reconstruction of this unknown tree.

In cases where the cell division rates are difficult to determine, simulations allow us to estimate the expected levels of accuracy for a range of parameters. As we explain below (in response to point 5), we find that even when mutation rates cannot be matched precisely to cell division rates, lineage reconstruction can be quite accurate, given a sufficient number of targets.

Are some CRISPR recorder designs fundamentally better than others, and does this study suggest a better approach?

Our study suggests that recorder design is important, and that adjusting target number, mutation rate and dropout rates will each have a major impact for lineage reconstruction, with any type of recorder. This message is explicit throughout the manuscript.

Different recorder implementations (GESTALT, MEMOIR, ScarTrace, LINNAEUS…) are likely to differ in performance, and this will largely depend on specific target numbers, mutation rates, etc. In this manuscript we specifically test the performance of two published recorders (GESTALT and MEMOIR). We suggest that simulations would be valuable for testing any recorder design.

Major comments:1) Recently, there was a systematic follow up to van Overbeek et al., 2016, by Leopold Parts at the Sanger (Allen BioRxiv 2018), which used a large dataset to provide an indel distribution prediction tool. It would be very interesting to consider this for specific target sequences, and discuss "optimized" spacer sequences for CRISPR recorders that have higher mutational diversity than others.

Indeed, the study of Allen et al. (and two other recently published studies, by Shen et al. and Chen et al.) suggests that the diversity and relative frequencies of mutational outcomes are influenced by the target and flanking sequences. We agree that this will be important to consider when designing recorder sequences. These predictions will need to be validated experimentally in the relevant species and in the context of the genomic locus harbouring the CRISPR recorder.

In the revised text we acknowledge this as follows: “If, as expected, the diversity of mutations and their relative frequencies vary depending on the target sequence and its local environment (Overbeek et al., 2016; Vu et al., 2017; Allen et al., 2018; Shen et al., 2018; Chen et al., 2018) sampling different targets to approach this optimum would be worthwhile.”

2) The rationale for assuming that CRISPR recorders generate irreversible target mutations is unclear. For example, there can be a "back rate" where a mutated target that creates a single base indel can still be recognized by Cas9 after a cell division.

To our knowledge no systematic study has been carried out to address the efficiency of re-targeting of mutated sites by the same gRNA. We expect the frequency of reversions to be very low, however, because (1) following mutation, the majority of targets have multiple mismatches to the gRNA near the CRISPR cleavage site (Allen et al., 2018) and these sites are expected to be targeted with very low efficiency compared with unmutated targets (Hsu et al., 2013), (2) even single nucleotide mismatches near the cleavage site can be deleterious for CRISPR targeting (see Hsu et al., 2013 and our results), and (3) even when mutated targets are re-cleaved, the great majority will not revert to the unmutated sequence (see mutational outcomes in Allen et al., 2018). Based on these data, we estimate that reversion rates will be much lower than 1% of the forward mutation rates. For these reasons we consider that the rate of reversion is negligible. “Irreversibility” seems to be a reasonable approximation when simulating the mutational process and lineage reconstruction.

We have added a sentence to the Results to this effect: “We expect the frequency of reversions to be negligible as even single nucleotide changes result in a large decrease in mutation rate.”

3) Why is neighbor-joining chosen over parsimony (and over several other possible algorithms?) This should be elaborated on. Can the authors leverage their insights to improve these reconstruction algorithms to specifically address the challenges of CRISPR recorders?

The choice of NJ over Parsimony was a practical one. We have shown that a Parsimony approach would be slightly more accurate on average than NJ, however it is considerably slower. We have now quantified the difference in speed. In the revised manuscript we show that the time taken by the parsimony method increases exponentially with lineage depth, such that reconstructing a lineage of just 180 cells takes ~52 hours (Figure 2—figure supplement 1B). With NJ the same tree can be reconstructed in less than 1 second.

While it may be worth attempting to use Parsimony with a real data set (one would probably have to take a two-step heuristic approach and produce an initial best guess tree using NJ, followed by optimisation using the parsimony criterion and branch swapping), for our purposes of comparing thousands of different simulations it is important to have a good method and essential to have one that is computationally tractable. Most of our analysis concerns comparisons of the effects of different designs and various parameters on the accuracy of lineage reconstruction, and the conclusions hold as long as the tree reconstruction method is consistent.

In the revised manuscript we address this point as follows:

1)We provide a more detailed comparison of the accuracy of NJ and Parsimony methods (Figure 2—figure supplement 1A and B). We find that Parsimony performs 5-10% better than NJ for trees of up to 180 sampled cells.

2) We justify the choice of NJ in the Materials and methods section.

4) The chosen definition of "accuracy" seems to be problematic because it does not generally discriminate between false positive and false negative reconstruction events. This is briefly considered in Figure 7 but should be expanded.

Robinson-Foulds is a standard metric that is widely used to compare trees in phylogenetics. We adopted a metric related to Robinson Foulds that gives the percentage of correctly resolved splits as a global measure of tree accuracy. This is suitable for comparing the overall accuracy of lineage reconstruction achieved in different conditions. This has been clarified in the manuscript.

As suggested, we have expanded the use of False Positives and False Negatives – an alternative measure of tree accuracy intended to be more easily interpreted in a developmental context. Our original use of False Positives and False Negatives considered clones of just one size. We have now extended the FP/FN approach to encompass multiple different sized clones and have applied this extended approach to the simulation of the 32-target CRISPR recorder with optimised mutation rates with and without dropouts, to GESTALT and to MEMOIR (Section “Measuring accuracy at different depths of the tree” and Figure 9 in revised manuscript).

Finally, to provide a more complete estimate of accuracy at different depths of the lineage tree, we now record how the ability to reconstruct the cell lineage varies through the tree as the process of mutation accumulation proceeds (see “quartet analysis”). We present this analysis for the simulation of the 32-target CRISPR recorder with optimised mutation rates with and without dropouts, for GESTALT and for MEMOIR (Figure 8 in revised manuscript).

5) The authors show that setting the correct mutation rate matters. How can mutation rates be matched to the rate of cell division when the rate or interval of cell divisions is unknown/itself variable (e.g. in a tumor)? This would be worth discussing.

One of the important outcomes of our simulations has been to show that the range of mutation rates that produce similarly accurate lineage reconstruction can be quite broad (see Figure 2C,D). Thus, even when mutation rates cannot be matched precisely to cell division rates (such as when these are unknown or vary within a tree), a mismatch between mutation rates and expected rates of cell division will have relatively limited impact on overall accuracy within fairly broad limits.

We have added to the text as follows, to make this more clear: “The range of mutation rates that can produce accurate lineage reconstruction fortunately proves to be quite broad for a given tree size; 0.05 to 0.25 mutations per cell division can yield reasonably high levels of accuracy for trees of ~65,000 cells, if the division rates are relatively even (Figure 2C,D). This flexibility will be beneficial in cases where the cell division rates are poorly characterised.“

Reviewer #2:

The manuscript "Is it possible to reconstruct an accurate cell lineage using CRISPR recorders?" uses simulations and some experimental data to conduct a thorough exploration of the parameters governing the accuracy of CRISPR lineage tracing. They define and sweep four key parameters for two types of CRISPR recorders: number of targets, mutation rate, mutational character states, and dropouts. This work is important and valuable to a growing community exploring the potential of CRISPR recorders. It provides a path for future optimization of these type of lineage tracing tools. For the most part, the writing and figures are clear and informative. I have five major concerns listed below.1) The provocative title in the form of a question is somewhat misleading – is the answer yes? The manuscript doesn't explicitly answer this question. I would suggest a more accurate title, for example "design specifications for more accurate CRISPR recorders."

We think that the title of the manuscript (“Is it possible to reconstruct an accurate cell lineage using CRISPR recorders?”) accurately describes the question that we are addressing in this work. The answer to that question is not a simple yes/no response, but depends on the size and complexity of a lineage, on the design of the recorder, and on the level of accuracy required. Our paper addresses these points.

2) I have two concerns with the term "accuracy."First concern: The authors should be explicit upfront about their definition of the term accuracy, which includes both false positives and false negatives. I would argue that false positives (wrong branchpoints) are worse than false negatives (i.e. missing branchpoints). Can't an "accurate" tree have missing branchpoints but no wrong branchpoints? The authors briefly describe this distinction, but only at the very end of the Results section and then calculate these subscores of "accuracy" only for the SOLID sequencing approach. I would appreciate a longer discussion of this definition and scoring at the beginning of the results, and FP/FN calculated and reported for all simulations.

Please see above our response on this to reviewer 1. As explained in the Materials and methods section and depicted in Figure 9—figure supplement 1, both FP and FN are derived from a mismatch between the true tree and the reconstructed tree. The false negative cells are not missing from the tree, but missing from the reconstructed clade/clone to which they belong (they are incorrectly placed elsewhere in the tree).

Second concern: Generally speaking, how accurate do we need CRISPR recorders to be? The authors set a high bar: complete and accurate lineage tracing of a 65,000 cell tree. Is 4% or 14% (the final "scores" given to MEMOIR and GESTALT) good enough for many scientific applications? It is certainly better than nearly all existing lineage tracing techniques, from which we have learned a great deal about biology. I would appreciate a discussion of why 100% complete and accurate trees are such an important goal, and what we can still learn from less accurate trees.

We agree that different lineaging accuracies are required to address different problems. We discuss and respond to this point at length in our response to reviewer 1 (above).

3) I am concerned about three library prep or filtering steps in the sequencing of the Drosophila target site array that are not clearly explained in the text, which may reduce the number of character states used in all following simulations:a) To call character states – why only use 9bp flanking the target site instead of simply aligning each read to unedited sites? I would assume that this would eliminate some character states.

Indeed, mutations sometimes extend beyond the 9 bp window of detection, so it is important to ask whether a larger window would be more useful. As pointed out by reviewer 3, the trade-off for increasing the detection window is that more sequencing errors would be captured and counted as false character states.

We considered two points in opting for a 9 bp window. First, the nature of CRISPR-induced mutations is such that mutations are centered at the cleavage site. We estimate that mutations falling outside of this detection window have a lower frequency than the sequencing error rate (<0.4% of mutations are missed with a 9 bp window). Second, the mutations that extend beyond 9 bp are deletions (and a few insertions) that would typically bring unique sequences into the 9 bp detection window (see Allen et al. 2018). We expect that all these changes will be detectable as distinct mutations.

As the reviewer suggests, when we consider only 9 nucleotides we cannot distinguish certain different mutated outcomes from each other. This could add to the level of homoplasy and reduce accuracy.

Our analysis of the mutations captured when reading 18 bp as opposed to 9 bp showed we are incorrectly grouping 4 pairs of character states. Reading 18 bp, however, introduces additional sequencing errors: by reading 18 bp the sequencing error would increase from 1.1% to 2.2%.

To quantify the overall effect on lineaging accuracy, we compared the accuracy of lineage reconstruction using 9 bp to what we would obtain using 18bp (both coded as 60 character states). The mean accuracy improves to an insignificant degree, from 72.19% to 72.24%.

These results are included in the revised manuscript, in Figure 4—figure supplement 1.

b) If I understand Figure 4A, the authors use a primer that sits directly on the PAM -mutational outcomes that disrupt any bases within the PAM presumably would not be captured by the PCR and sequencing. Can the authors discuss the impact of this?

The targets were amplified and sequenced using primers located >40 nucleotides away from each target site. The ‘primer’ indicated in Figure 4A is incorrect and has been removed. We apologise for this mistake.

In section “DNA extraction, generation of libraries and sequencing” of the Materials and methods, we explain more clearly the use of amplification and sequencing primers.

c) Why merge 140 rare character states into a shared state (state 60)? This would obviously lead to tree errors. Why not treat these just as all other character states? The authors write that this was "for convenience," but I'm not sure why this is convenient and it seems to be a potential source of false positives.I believe other papers have shown a higher number of potential character states, and I'm concerned these steps listed above may impact the simulations.

As we mention in the section “Reconstructing lineage trees using Neighbor Joining (PAUP*)” of Materials and methods, the maximum number of distinct character states that we could use in the analysis programme PAUP* was 64.

We can show that there are diminishing returns of using a larger diversity of character states; for example, compare the very similar results when considering 16 versus 32 character states in Figure 2C. We have tested this further by comparing our results with 60 mutated character states to the case where we reduce the data to 40 character states (gathering together the lowest frequency outcomes). The accuracy of the optimal reconstruction changes from 72% to 71% (Figure 4—figure supplement 1).

4) The authors, as they admit, use "the most pessimistic estimate" of the frequency of dropouts. It is entirely possible to have two cuts within a cell cycle result in two edits instead of a dropout. Have the authors considered explicitly using their own experimental data (even with library prep caveats), or the GESTALT data, to simulate dropout rates? Since this assumption dramatically impacts accuracy, it would seem important to be careful about how to model dropouts.

We have simulated GESTALT based on the experimental data. When we model the worst case scenario for dropouts we observe a very similar number of alleles to the experimental data suggesting that reality approximates the worst case (see Figure 6—figure supplement 1).

For *Drosophila*, we do not have experimental data regarding dropouts, but we provide results covering both best and worst case scenarios.

5) Choice of tree reconstruction algorithm matters. In their own simulations, Parsimony appears more successful that Neighbor Joining (Figure 2—figure supplement 1). However, this was never stated explicitly in the text, and there is no discussion of algorithm selection and its impact on accuracy. While NJ was selected for obvious reasons (speed), the authors should provide a clear discussion of other options and their impact on tree accuracy.

We address this point in our response to reviewer 1. We have added new results in Figure 2—figure supplement 1A and B and an explicit statement to the Materials and methods section.

Minor Comments:1) GESTALT cell culture approach also used a similar strategy of an array of off-targets and 1 guide RNA, and should be cited when discussing the fly array design, along with the caveats associated with this approach (poor editing at many off targets).

We have added this citation in the revised manuscript:”We chose to adjust the mutation rate by altering the target sequence in order to introduce mismatches in the sgRNA:target pairing (similar to the strategy used on cultured cells by McKenna et al., 2016)”.

We do not consider poor editing rates at some target variants to be a problem. The purpose of this array was to measure the actual mutation rate on diverse targets, which enabled us to select targets that match the optimal rates for lineage reconstruction. The actual recorders include arrays of the selected target, and no suboptimal targets.

2) The MEMOIR paper extensively considers accuracy of tree reconstruction, including comparisons to reference trees. These data and discussions should be mentioned and referenced in this paper.

We thank the reviewer for pointing this out. We have included references to Frieda et al., 2017 (MEMOIR), Schmidt et al., 2017 and Spanjaard et al., 2018 in the revised manuscript (Introduction paragraph five).

Reviewer #3:

In this manuscript, Salvador-Martínez and Grillo et al. present a simulation study of newly developed CRISPR lineage tracing technologies. The authors do a good job of setting up the problem, explaining their choices of various parameters and assumptions, and adding experimental data in Drosophila to reinforce these choices. This work will be a valuable addition to a quickly advancing field, particularly as a reality check on the extent of organismal engineering that will likely be required to achieve near-complete, accurate trees by this class of methods. Our first major comment is about the tone of the paper, while our our additional major comments primarily relate to alternative measures that should be evaluated that strike more of a balance between strict accuracy and the general conservation of tree topology. Additionally, some basic flaws in the experimental design and analysis of editing outcomes in Drosophila should be addressed before publication.1) A first major comment is that the measure of accuracy used throughout the paper is very conservative, and more generally the tone that is struck in many parts of the paper is (we feel) overly conservative. Although strict accuracy and completeness are of course goals worth shooting for, they are not prerequisites these kinds of experiments to achieve biological insights. For example, lineage relationships between cell types might well be accurately inferred from the general topology of a tree that contained inaccuracies or uncertainty near its tips.

As discussed in our response to reviewer 1, we have tried to be clearer regarding the fact that the required level of accuracy will depend on the application.

One of the most important messages from our study is the desirability of performing such power analyses as a tool for optimising the design of the experiment and in order to have an estimate how reliable is the final result.

For any new technology, proof-of-concept studies are just that -- proof-of-concept, and it's always been clear that significant additional engineering would be (and still is) required to maximize the value of these methods. This point does not detract from the value of the simulations presented in this paper. A more optimistic take on the same results is that it is possible to reconstruct large trees with reasonably high accuracy (great!), but it will require the introduction of at least 50 targets (and ideally several hundred targets), tuning of the mutation rate (although the broad plateau presented in Figure 2D is rather encouraging), and careful consideration of variable cell division rates. These conclusions and other analyses presented in the paper provide important guidance for the field (and a reality check against short term thinking), but the paper often slips into a negative tone that in our view is inconsistent with the results themselves (e.g. the fact that conditions are identified where reconstruction with 99% accuracy is achieved; subsection “Optimising cell lineage reconstruction for in situ sequencing with 2, 4 or 16 character states” paragraph six). We urge the authors to: (a) make it clearer, from the beginning of the paper, the extent to which accuracy as defined here is a highly conservative definition, relative to what might be required to achieve biological insights from trees reconstructed from GESTALT or related methods.

Please see our response to reviewer 1.

(b) strike a more balanced tone, with less emphasis on what is not possible using the systems as reported in their proof-of-concept implementations, and more emphasis on the path forward, i.e. the extent to which further engineering (more targets, reducing inter-site deletion, tuning of mutation rate, etc.) is required to get the most out of these methods.

The manuscript assesses the accuracy of current methods without commenting on what is possible or not possible to achieve. We explicitly discuss the ways in which CRISPR recorders can be improved and quantify the expected improvements in performance. We conclude that CRISPR-based methods should ultimately allow us generate accurate lineage trees.

The last paragraph of the manuscript has been revised to give a clear positive answer to the question posed in the title: “This approach should stimulate the development of a new generation of CRISPR recorders whose performance will be assessed explicitly and adapted to match the biological questions at hand. Our study predicts that optimized recorders will ultimately allow the reconstruction of accurate cell lineages of complex multicellular organisms at the level of a single cell.”

2) The simulation of CRISPR-based lineage tracing technologies in the first section (Figure 2) focuses on accuracy with Robinson-Foulds, but should include characterization with the clonal reconstruction measurements (FP / FN analysis). These measurements are used later in the MEMOIR simulation section, and many people will be interested in this use of lineage tracing technologies for this purpose. A mention or comparison to other distance metrics might be more appropriate, see https://www.ncbi.nlm.nih.gov/pubmed/21383415 or the review https://www.ncbi.nlm.nih.gov/pubmed/25378436

As discussed in response to referees 1 and 2 we have now expanded our use of False Positive and False Negatives. While other measures of tree similarity exist we expect that their use would not change our conclusions as to the optimal value for a given parameter.

3) You show higher accuracy with maximum parsimony approaches, but use neighbor-joining throughout the paper. A strong justification of this choice is important, as this appears to be a strong bias against the methods you're evaluating.

We have addressed this in our response to referee 1 (point 3).

4) Robinson-Foulds is a distance metric, and here you've normalized it to an accuracy over [0-100%], which is not detailed anywhere in the paper. Details in the Materials and methods section would improve the clarity of the paper.

We have now made clearer that our measure is related to but distinct from the Robinson Foulds distance metric. We state: “The accuracy of lineage reconstruction of each simulation was determined by comparing the inferred tree with the reference tree using a measure derived from the Robinson-Foulds algorithm (Robinson and Foulds, 1981), which calculates the percentage of splits in the reference tree that are precisely recovered in the inferred tree (Figure 2B). If the inferred tree is identical to the reference tree, the Robinson-Foulds accuracy is 100%.”

And in the Materials and methods section we state: “The accuracy of each cell-lineage reconstruction was determined by calculating a measure related to the Robinson-Foulds distance (RF) between the reference and the inferred trees. We count the percentage of splits (sets of cells separated into two groups by a node in the tree) in the reference tree that are also found in the inferred tree.”

5) Counting 9mers is a bit liberal for determining editing outcomes from the FAST target sequencing. Given the known double-stranded break location and repair outcomes, most mutational outcomes should be centered at, or overlap the cutsite. By including all of the 9 proximal bases, the captured editing diversity will include sequencing errors and PCR errors. This shouldn't affect the overall profile of editing outcomes, but will increase the number of mutations, and should be mentioned. This is made apparent by the elevated mutation rates in the untargered column of Supplementary Table 1 in Supplementary file 1, as some of the more active off-targets have changes in bases believed to strongly obstruct Cas9 binding (bases that are very close the PAM sequence).

As we acknowledge in our response to reviewer 2 (point 3a), there is a trade-off in setting the size of the detection window, between capturing mutational diversity and excluding sequencing errors.

We have directly quantified the PCR/sequencing error rate in flies that did not carry Cas9 (shown as “untargeted” in Supplementary Table 1 in Supplementary file 1). For the FAST target, we find this rate to be 1%, compared with CRISPR target mutagenesis of 87% in embryos and 92% in adults.

We have amended the following text in the Results section to make this more explicit: “As expected, the target that has perfect complementarity with the Emx1.6 sgRNA (target 16, named the “FAST” target) showed the highest mutation rate; having corrected for sequencing errors (~1% of control targets have differences due to PCR or sequencing errors, Supplementary file 1) we observed that 87% of the targets carried a mutation at the end of embryogenesis.”

6) Also the location of the primer (Figure 4) precludes find deletions that extend downstream into the PAM sequence. This eliminates the detection of mutations that extend 3' of the cutsite, and will deflate the diversity of editing you see. This is a major bias, and could also affect the FAST off-target results.

As indicated in our response to reviewer 2 (point 3b), there was a mistake in Figure 4, which we have now corrected. The targets were amplified and sequenced using primers located >40 nucleotides away from each target site so we were able to find all deletions rather than only those in the 9bp window.

7) It's been shown that very large deletion frequency decreases with the distance between two cutsites (https://www.ncbi.nlm.nih.gov/pubmed/24907273). it would be worth including in the dropout simulations, or at least mentioning an alternative model with decreasing dropout efficiency at larger distances.

Indeed, besides the rate of cleavage by CRISPR, dropout frequency will be influenced by the interplay of several parameters – such as the distance between cut sites, the speed of repair, etc. How these parameters influence dropout rates is currently difficult to predict and is likely to vary across species and cell types. The data presented in the cited paper refers to variations in deletion rates over large distances (multiple kb) in mammalian cells. It does not seem particularly relevant for the CRISPR recorders that we simulate, which are much smaller in size (inter-target distances of 100 bp for our FAST target arrays and 26 bp for GESTALT).

Because dropout rates are difficult to predict, our analysis of 32 targets covered two conditions: a best case scenario with no dropouts and a worst case scenario with dropouts for every predicted double cleavage. For GESTALT the worst case scenario recapitulates the experimental data available.

8) It would be good to characterize the accuracy of genotype collapsed trees (https://www.ncbi.nlm.nih.gov/pubmed/29474671) which aim to reduce the number of false branch points (branch points introduced by the tree bifurcation requirement, not mutations).

This is an interesting idea and one of several that deserve to be considered in the future for accurate reconstruction of lineage data but we feel that this is outside the scope of the current manuscript.

9) In the SOLiD simulation section (for Figure 7) it's unclear what sequence would be used for the primer, and how often that binding sequence would be obstructed by deletions in its target sequence. Some more details here would be helpful.

In our SOLiD simulations we assumed that the sequencing primer would never be lost by mutation, so our simulation represents a best-case scenario. To minimise homoplasy when capturing one or two SOLiD reads, we found that the primer should be optimally placed on the 3’ of the target, overlapping the PAM.

This experiment simulates the additional constraints imposed by an existing technology for in situ sequencing and highlights the need to consider factors that can affect the quantity and quality of information coming out of the experiment.

We have clarified the SOLiD experiment by adding the following sentences in Materials and methods: “For the simulation of SOLiD sequencing we used the real distribution of the dinucleotides observed at positions 6-7 and 11-12 upstream of the PAM. We translated these into four different colour states per dinucleotide according to the SOLiD protocol (Figure 7—figure supplement 1).”

Minor Comments:Fourth sentence of the Abstract should probably say approaches (you profile at least two).

Changed to “Here, we use computer simulations to estimate the performance of these approaches under different conditions.”

Abstract: not all terminal branches are fully differentiated cells.

Changed to “The divisions that generate these adult cells constitute a genealogical tree with the fertilised egg at its root and each adult cell as a terminal branch.“

Introduction paragraph two: there's probably a better phase than “simple cases” (it was quite the effort, which you mention further on).

Changed to “Obtaining high resolution (single-cell level) lineages is a challenging task that has been solved only in animals with relatively few cells, such as the nematode *Caenorhabditis elegans*: its complete lineage (∼1000 cells) was deduced by painstaking observation of each cell division under the microscope.”

Introduction paragraph eight is a bit strong. Certainly simulations will inform future synthetic recording systems, but the assumptions and simplifications of simulation might prevent finding the optimal solution without more true biological validation.

Changed to “Ultimately, these simulations will help us to establish a set of criteria for the optimal design of CRISPR-based lineage recorders, as well as to understand the limitations of these techniques when addressing real biological questions.”

Subsection “Optimising cell lineage reconstruction for in situ sequencing with 2, 4 or 16 character states” paragraph nine: shouldn't "double the number of reads per target" be "double the read length" or "double the number of cycles"?

Changed to “Clearly, increasing the number of targets will improve performance, but we wanted to know whether it would be better instead to read double the number of nucleotides per target, which represents the same sequencing effort.”

Really beautiful figures all through the supplement, but the most minor change, add " + theme_classic()" or similar to the R command for Figure 2—figure supplement 1.

The figure has been revised.

Additional details are needed in the experimental section about the PCR reaction.

We have added more details to address this point.